# Diversifying crop rotation increases food production, reduces net greenhouse gas emissions and improves soil health

Xiaolin Yang [1,2] ✉, Jinran Xiong[1,2], Taisheng Du [1,2] ✉, Xiaotang Ju [3] ✉, Yantai Gan [4,5] ✉, Sien Li[1,2], Longlong Xia [6], Yanjun Shen [7], Steven Pacenka[8], Tammo S. Steenhuis[8], Kadambot H. M. Siddique [9], Shaozhong Kang [1,2] & Klaus Butterbach-Bahl[10,11]

Global food production faces challenges in balancing the need for increased yields with environmental sustainability. This study presents a six-year field experiment in the North China Plain, demonstrating the benefits of diversifying traditional cereal monoculture (wheat–maize) with cash crops (sweet potato) and legumes (peanut and soybean). The diversified rotations increase equivalent yield by up to 38%, reduce $N_2O$ emissions by 39%, and improve the system's greenhouse gas balance by 88%. Furthermore, including legumes in crop rotations stimulates soil microbial activities, increases soil organic carbon stocks by 8%, and enhances soil health (indexed with the selected soil physiochemical and biological properties) by 45%. The large-scale adoption of diversified cropping systems in the North China Plain could increase cereal production by 32% when wheat–maize follows alternative crops in rotation and farmer income by 20% while benefiting the environment. This study provides an example of sustainable food production practices, emphasizing the significance of crop diversification for long-term agricultural resilience and soil health.

Producing more nutritious food to alleviate world hunger[1] while safeguarding the environment[2–4] is a significant challenge for humanity. The challenge is much pronounced in highly-populated countries and regions where agricultural resources are limited. Conventional intensified food production systems' reliance on major inputs of synthetic agrochemicals, such as fertilizers and pesticides, have boosted food production significantly since the 'Green Revolution', but these

systems have also emitted great amounts of greenhouse gases (GHG) and caused environmental degradation[4–6]. For instance, China's crop production has increased by 74% in the past 30 years (1986–2016), but this came with a >300% increase in synthetic fertilizer use[7]. In 2019, China's GHG emissions associated with food production reached 2.4 Gt $CO_2$-equivalent ($CO_2$-eq)[8], with about 50% emitted during the crop production stage[8,9]. The loss of soil fertility which may go along

[1]State Key Laboratory of Efficient Utilization of Agricultural Water Resources, Beijing 100083, China. [2]College of Water Resources & Civil Engineering, China Agricultural University, Beijing 100083, China. [3]School of Tropical Agriculture and Forestry, Hainan University, Haikou 570228, China. [4]College of Life and Environmental Sciences, Wenzhou University, Wenzhou, Zhejiang 325035, China. [5]The µBC-Soil Group, Tallus Heights, Kelowna, BC, Canada. [6]State Key Laboratory of Soil and Sustainable Agriculture, Institute of Soil Science, Chinese Academy of Sciences, Nanjing 210008, China. [7]Key Laboratory of Agricultural Water Resources, Centre for Agricultural Resources Research, Institute of Genetics and Developmental Biology, Chinese Academy of Sciences, Shijiazhuang 050022, China. [8]Department of Biological and Environmental Engineering, Riley-Robb Hall, Cornell University, Ithaca, NY 14853, USA. [9]The UWA Institute of Agriculture, The University of Western Australia, Perth, WA 6001, Australia. [10]Land-CRAFT, Department of Agroecology, Aarhus University, Aarhus, Denmark. [11]Institute of Meteorology and Climate Research, Atmospheric Environmental Research (IMK-IFU), Karlsruhe Institute of Technology (KIT), Garmisch Partenkirchen, Germany. ✉e-mail: yangxiaolin429@cau.edu.cn; dutaisheng@cau.edu.cn; juxt@cau.edu.cn; gary.gan@wzu.edu.cn

with the intensification of crop production further complicates food production[10–12] and exposes it to climate risks[13,14] and environmental health concerns[15,16].

Innovative concepts like integrated farming systems with diversified crop rotations[17,18] have emerged to address these food production and environmental sustainability challenges. These systems offer a range of food crops to meet consumer demand for plant-based, healthy food[19]—an increasing dietary trend in high and upper-middle-income countries[20]—while providing other agricultural products such as animal feed, industrial fiber, or multi-purpose biofuels[21,22]. Moreover, integrated food production systems help increase farmers' income and deliver socio-economic benefits[20,23]. However, little is known about how cash-crop and legume-diversified cropping systems can achieve the triple goals—increasing food yields, reducing environmental footprint, and benefiting soil health.

In this context, we conducted a 6-year (2016–2022) field study in the North China Plain—the food basket of China—one of the most intensively cultivated regions in the world, where crop production is dominated by simple winter wheat (*Triticum aestivum* L.) and summer maize (*Zea mays* L.) double cropping (wheat–maize, WM; two crops 1 year) which occupies 70% of the area's arable land, delivering about 23% of China's total cereal food[24]. The specific objective of the case study was to asses comprehensively several diversified cropping systems in terms of food production, GHG balance, soil health benefits, and farmers' income. We tested various diversified cropping systems, incorporating cash-crops, legumes, other cereals, or forages into the conventional wheat–maize system. Major performance metrics investigated were: (i) plant biomass, grain and protein yields, and farmers' net incomes; (ii) soil GHG fluxes, soil carbon (C) sequestration, and net GHG emissions; and (iii) soil health parameters including soil pH, bulk density, soil water content, total nitrogen (TN), dissolved organic carbon (DOC), soil nitrate-N ($NO_3^-$-N), ammonium-N ($NH_4^+$-N),

available phosphorus (AP), microbial biomass carbon (MBC), microbial biomass nitrogen (MBN), and microbial community composition and diversity. These measurements enabled us to test the hypotheses that (a) cash-crop diversified system increases farmers' net income without jeopardizing crop yields, (b) legume diversified systems reduce field scale GHG emissions, and (c) integrating diversified rotations increases food production, reduces GHG emissions, and benefits soil health (Fig. 1). Our results demonstrate that (a) instructive findings from newly designed, tested, and validated diversified systems could guide the North China Plain in establishing a more sustainable system to maintain or increase grain and protein production with reducing the damage to the environment and soil ecosystems, and (b) the results from such a representatively intensive food producing region may provide a guide for the countries/regions with similar agricultural environments to follow on an expanded scale.

## Results

### Diversified crop rotations increase ecosystem productivity

The food crops evaluated in the study are from different genera families; thus, the yield of each crop was converted to the 'equivalent' product yield to wheat for a compatible comparison (detailed in Methods). The sweet potato (*Ipomoea batatas* L.) →winter wheat–summer maize rotation (SpWM) increased the annual equivalent yield by 38% compared to the conventional winter wheat–summer maize double-cropping which yielded 13,185 kg ha⁻¹ annually (Fig. 2a). Rotations diversified with sweet potato (SpWM), peanut (*Arachis hypogea* L.) (PWM), or soybean (*Glycine max* L.) (SWM) significantly increased the annual economic benefit (net income) compared to the winter wheat–summer maize control (Fig. 2b), with sweet potato→winter wheat–summer maize rotation increasing 60%, and peanut→winter wheat–summer maize rotation and soybean→winter wheat–summer maize rotation increasing by 13–22% (*P < 0.05*). Protein

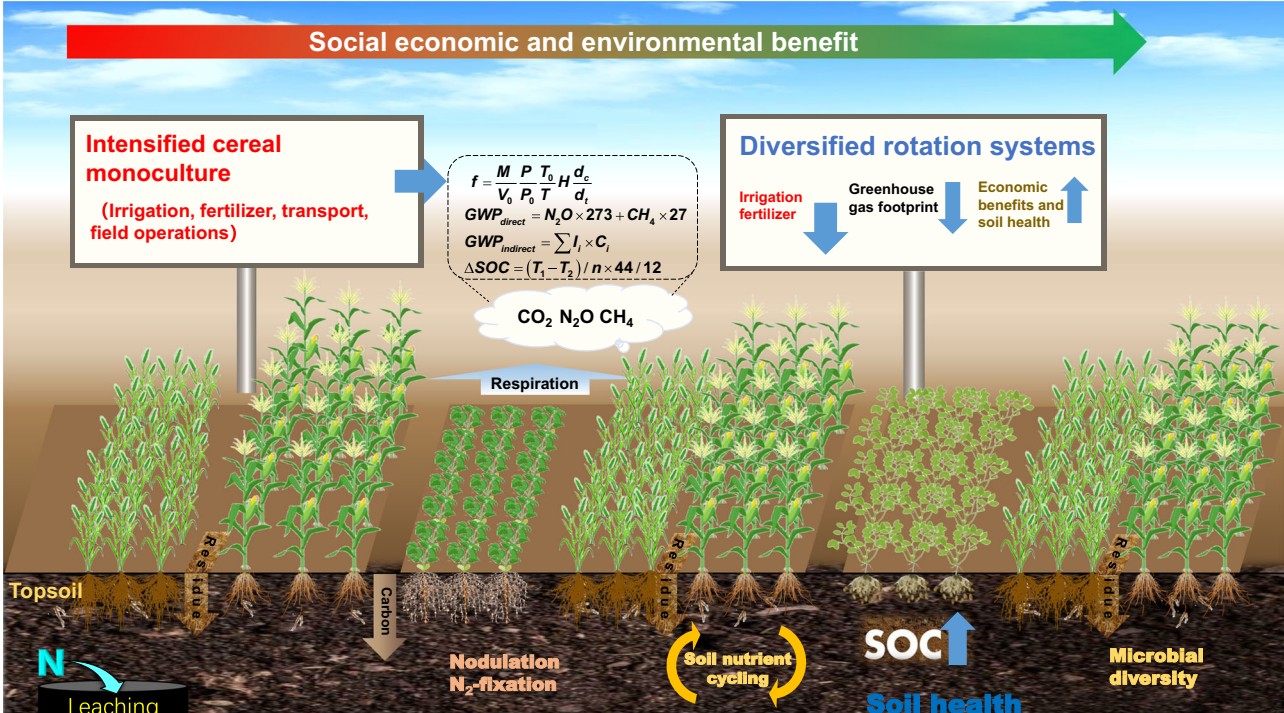

**Fig. 1 | Schematic illustration of system integration from issues to outcomes.** In the North China Plain—the case study area, traditional cereal monoculture (such as wheat–maize double-cropping, i.e., two cereal crops per year) requires inputs of synthetic agrichemicals and irrigation in food production, causing large greenhouse gas (GHG) emissions; in contrast, rotation systems diversified with cash and legume crops can maintain crop yields, increase farmers' income, and reduce GHG emissions due to the biological $N_2$ fixation by legumes partly substituting for synthetic N inputs. Legume-included rotations can also enhance soil health by stimulating soil microbial activities, increasing carbon sequestration, and enhancing nutrient cycles.

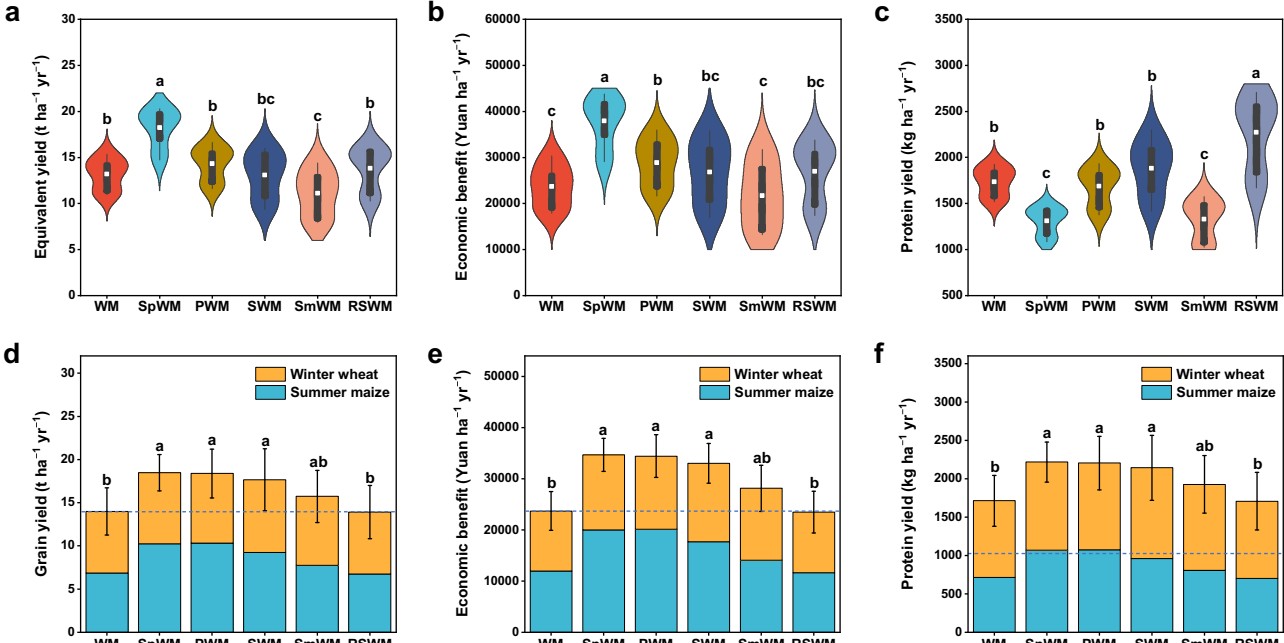

**Fig. 2 | Agroecosystem productivity.** The entire rotation system productivity: **a** equivalent yield to wheat, **b** economic benefit – net income, and **c** protein yield. The productivity of WM preceded by alternative crops in rotation: **d** annual grain yield, **e** economic benefit, and **f** protein yield. In **a–c**, the black bar within violin plots shows the 25 and 75 percentiles, the whiskers beyond the bar represent 95 and 5 percentiles, and the inner white dot indicates the median. The violin shaped area reveals the data distribution. The dashed lines in **d–f** are the baseline values of the WM rotation, and the error bars are the standard deviation of annual value summed over winter wheat and summer maize. In **a–f**, one-way ANOVA with two-sided and post-hoc test was conducted to determine significant differences. Different lowercase letters denote significant differences between the rotations at $P < 0.05$. The exact $P$ values: $P < 0.001$ in **a–c**, $P = 0.001$ in **d**, $P = 0.032$ in **e** and $P = 0.004$ in **f**. In **b** and **e**, \$1 US = 6.95 Chinese Yuan (as of May, 2023). For **a–c**, $n = 9$; **d–f**, $n = 18$. Treatment abbreviations: WM winter wheat–summer maize (control case), SpWM sweet potato→winter wheat–summer maize rotation, PWM peanut→winter wheat–summer maize rotation, SWM soybean→winter wheat–summer maize rotation, SmWM spring maize→winter wheat–summer maize rotation, RSWM ryegrass–sorghum→winter wheat–summer maize rotation. Source data are provided as a Source Data file.

yield varied among the rotation systems, with ryegrass (*Lolium multiflorum* L.)–sorghum (*Sorghum bicolor* L.) rotation (RSWM) and soybean→winter wheat–summer maize rotation having the highest protein yields at 2274 and 1883 kg ha⁻¹ yr⁻¹, respectively, being 8–31% higher than winter wheat–summer maize rotation (Fig. 2c). The ryegrass-sweet sorghum→winter wheat–summer maize rotation, with four crops in 2 years, produced the most aboveground biomass for livestock, while the soybean→winter wheat–summer maize rotation produced the most protein for human nutrition. In contrast, an entirely cereal-based system (i.e., spring maize→ winter wheat–summer maize rotation, SmWM) decreased equivalent yield by 16% and protein yield by 23% compared to the winter wheat–summer maize control.

The preceding crops in rotation significantly affected grain yield (Fig. 2d), net income (Fig. 2e), and protein yield (Fig. 2f) of the succeeding winter wheat–summer maize crops as the rotation continued. The diversified crop rotations with sweet potato, peanut, and soybean had a positive carryover effect on the productivity of the succeeding winter wheat and summer maize. On average, winter wheat–summer maize preceded by non-cereal crops in rotation showed increased grain yield, economic benefit, and protein yield by 26–32%, 39–46%, and 25–29%, respectively, as compared to the winter wheat–summer maize cereal monoculture.

**Diversified crop rotations reduce net GHG emissions**

The amounts of supplementary irrigation and fertilizers applied to each rotation varied among them, as did the N₂O and CH₄ emissions measured weekly from October 2016 to October 2022 (Supplementary Figs. S1–2). The diversified crop rotations significantly reduced annual cumulative N₂O emissions compared to the wheat–maize

control which was at 8.9 ± 1.0 kg N ha⁻¹ (Table S1, Fig. 3a), decreasing by 30%, 42%, and 49% in the peanut→wheat–maize rotation, soybean→wheat–maize rotation, and sweet potato→wheat–maize rotation, respectively. These rotations with peanut, soybean, and sweet potato received a reduced dosage of N fertilizer as compared to the wheat–maize control (Table S1). The soils of all crop rotations, including wheat–maize, acted as net sinks for atmospheric CH₄. However, the diversified crop rotations increased the sink strength by 33–76% above wheat–maize (Fig. 3b). The wheat–maize control had the highest annual global warming potential (GWP) for soil N₂O and CH₄ fluxes (3764 kg CO₂-eq ha⁻¹ yr⁻¹, Fig. 3c). The cereal-including rotations (ryegrass-sweet sorghum→wheat–maize rotation and spring maize→wheat–maize rotation) had 22% and 19%, respectively, lower GWPs than wheat–maize rotation, while the rotations including legumes (peanut, soybean) or sweet potato had 32%, 43%, and 51%, respectively, lower GWPs than wheat–maize rotation. Besides field-scale direct GHG emissions, indirect GHG emissions associated with the use of agrochemicals and irrigation were highest for wheat–maize rotation (8802 kg CO₂-eq ha⁻¹ yr⁻¹) and 34–41% lower for rotations with sweet potato, peanut, soybean and spring maize (Fig. 3d).

The soil carbon stocks (SOC) in the 0–90 cm layer significantly increased from 2016 to 2022 for all treatments, but the magnitude of the change differed among rotation systems. Six years after the initiation of the experiment (in 2022), the peanut→wheat–maize rotation had the highest soil C sequestration (2.03 t ha⁻¹ yr⁻¹), followed by soybean→wheat–maize (1.91 t ha⁻¹ yr⁻¹) and sweet potato→wheat–maize (1.44 t ha⁻¹ yr⁻¹), while ryegrass-sweet sorghum→wheat–maize, spring maize→wheat–maize rotation, and wheat–maize rotations had significantly lower C sequestration rates (0.21–0.69 t C ha⁻¹ yr⁻¹) (Fig. 3e). Soil C sequestration offset total GHG

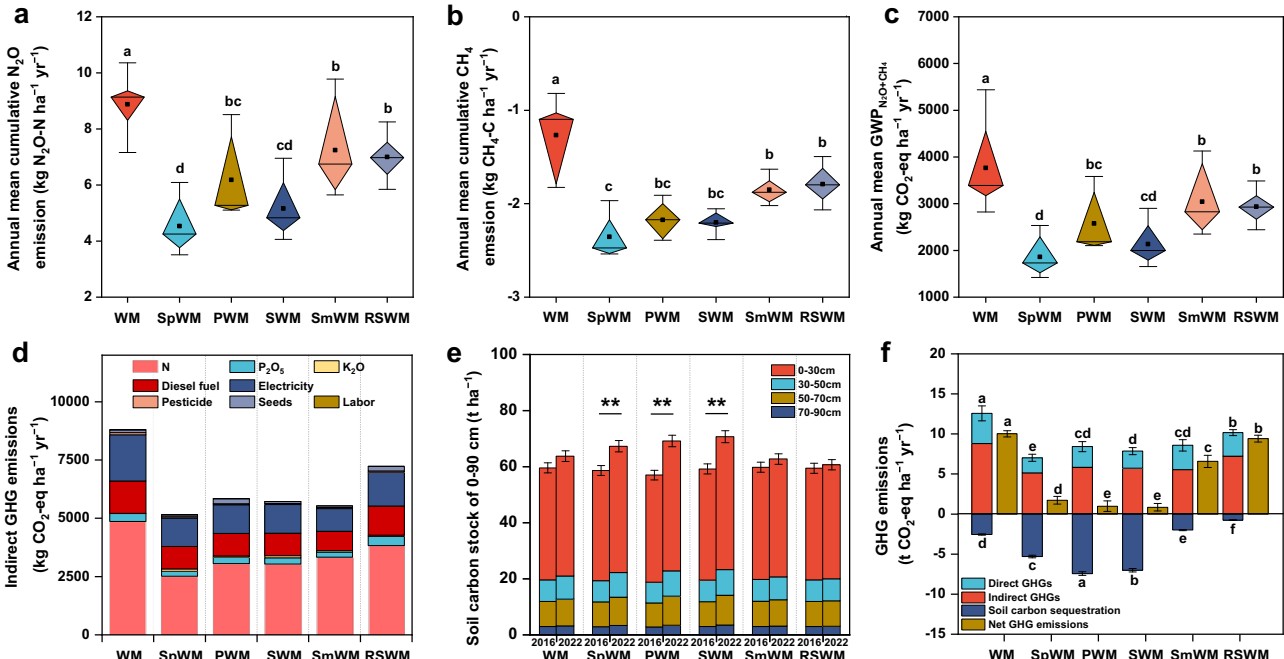

**Fig. 3 | Direct and indirect greenhouse gas (GHG) emissions in the diversified crop rotations. a** Annual mean N₂O emissions; **b** CH₄ emissions; **c** GWP$_{N2O+CH4}$; **d** indirect GHG emissions; **e** changes in soil carbon stock in the 0–90 cm depth; **f** net GHG emissions. In **a–c**, diamond box plots show triangles spanning 25 and 75 percentiles meeting at the median. Small squares indicate means. Whisker lines extend to the minimum and maximum. In **e** and **f**, error bars in each column represent the standard deviation of whole-profile sums of three replicates. In **a–c**, **e**, and **f**, one-way ANOVA with two-sided and post-hoc test was conducted for statistical significance; different lowercase letters denote significant differences between the rotations at $P < 0.05$. The exact $P$ values: $P < 0.001$ in **a–c** and **f**. In **e**, ** denotes significant changes in soil carbon stock in the 0–90 cm soil layer between 2016 and 2022 at $P < 0.01$ ($P = 0.005$ for SpWM, $P = 0.002$ for PWM and SWM). Treatment abbreviations: WM winter wheat–summer maize (control case), SpWM sweet potato→winter wheat–summer maize rotation, PWM peanut→winter wheat–summer maize rotation, SWM soybean→winter wheat–summer maize rotation, SmWM spring maize→winter wheat–summer maize rotation, RSWM ryegrass-sorghum→winter wheat–summer maize rotation. Source data are provided as a Source Data file.

emissions by 75–89% in the rotations with sweet potato, peanut, and soybean and 7–21% in the spring maize→wheat–maize, ryegrass-sweet sorghum→wheat–maize and wheat–maize rotations (Fig. 3f). Consequently, wheat–maize rotation had the largest net GHG emissions totaling 10,025 kg CO₂-eq ha⁻¹ yr⁻¹, and the diversified rotations with sweet potato, soybean and peanut lowered net GHG emissions by 83%, 90%, and 92%, respectively, as compared to wheat–maize ($P < 0.05$).

## Diversified crop rotations enhance soil health and microbial diversity

Soil health is often disregarded in the assessment of food crop production systems. In the present study, we used the Cornell Soil Health Assessment (CSHA) scoring method and applied Principal Component Analysis (PCA) to evaluate how crop diversification influenced soil health. A soil health score was calculated for each rotation based on key indicators (detailed in Methods). The PCA revealed that the first two principal components accounted for 70.5% of the cumulative percent variability (PC1 = 51.1% and PC2 = 19.4%) in soil health (Fig. 4a), with the measured soil health indicators clustered in distinct groups. At the end of the 6-year study, the peanut→wheat–maize rotation had the highest soil health score (59.5), followed by soybean→wheat–maize rotation (56.8) and sweet potato→wheat–maize rotation (52.9) (Fig. 4b), being 41–59% higher than wheat–maize (33.2).

A closer examination of the soil indicators that contributed to soil health scores revealed that the soil in the peanut→wheat–maize rotation had a 6.5% higher SOC concentration and 29.7% higher dissolved organic carbon (DOC) but 5.7% lower total N and 15.4% lower available P (AP) concentration than wheat–maize (Fig. S3). These changes were due to reduced N fertilizer (−41%) and associated higher nutrient use efficiencies by crops in the diversified rotations. The ryegrass-sweet

sorghum→wheat–maize rotation had the lowest SOC and DOC due to the removal of forage for silage during the growing season, which stimulated soil microbial activity, leading to the highest MBC (Fig. S3).

Soil microbial community composition and diversity—an indicator of the biological mechanism of soil health—were determined in 2016 and 2022 to evaluate the cumulative changes after 6 years' rotational experimentation. It showed that the diversification of crop rotations had positive effects (Fig. 5). For example, the Shannon index —a popular index for quantifying microorganism diversity taking into account the number of species and relative abundance of each species in a sample—significantly increased by 7–10% in the rotations with sweet potato, peanut, and soybean rotations, but did not change in wheat–maize or the graminaceous crop-based spring maize→wheat–maize and ryegrass-sweet sorghum→wheat–maize systems from 2016 to 2022 (Fig. 5a). Inclusion of the legumes peanut or soybean in rotations increased bacterial community operational taxonomic units (OTU) richness (Fig. S4a–f) and fungal community Shannon index, Chao 1, and OTU richness ($P < 0.05$) compared to wheat–maize (Fig. S4g–l). Many of the soil health indicators are dynamic and may change weekly, seasonally, or annually. However, after 6 years of 'rotation nourishment,' soils in the rotations with sweet potato, peanut, and soybean had significantly higher values for Chao 1, abundance-based coverage estimator (ACE), and community richness, compared to the cereal monoculture spring maize→wheat–maize system (Fig. 5b–d).

Redundancy analyses, depicting the association of soil physico-chemical properties and crop rotations in the bacterial (Fig. 5e) and fungal (Fig. 5f) communities, found that the first two sets of components explained 54.6% (bacterial) and 43.5% (fungal) of the total variation in the two communities, respectively. The relationship was

complex, but it was evident that bacterial and fungal community composition changed from 2016 to 2022. The rotations with sweet potato, peanut, and soybean had distinct bacterial community compositions from the other three systems (Fig. 5e), whereas the five studied diversified rotations had fungal community compositions distinctly different from the wheat–maize control (Fig. 5f). The 6 years of rotational nourishment affected soil physiochemical properties interactively with soil microbial communities. The rotations with sweet potato, peanut, and soybean bacterial communities were closely associated with SOC and AP, whereas those of wheat–maize, spring maize→wheat–maize, and ryegrass-sweet sorghum→wheat–maize rotations were closely associated with TN and microbial biomass carbon (MBC). In contrast, the rotations with sweet potato, peanut, and soybean fungal communities were associated with SOC, pH, and DOC.

### Multiple functions assessments of diversified crop rotations

We used the comprehensive evaluation index concept (CEI) to assess the synergies and trade-offs of the different crop rotations related to yield, nutritional value, soil-related indicators (health, C sequestration, microbial biodiversity), net GHG emissions, and economic benefit (Eqs. 11–20 in Methods) (Fig. 6a, d). The results showed that the wheat–maize rotation had the lowest CEI value, averaging 0.19 (Fig. 6b), while the sweet potato→wheat–maize rotation had the highest CEI value of 0.81, followed by peanut→wheat–maize rotation of 0.75, and soybean→wheat–maize rotation of 0.69. The diversified rotations with sweet potato, peanut, and soybean had CEI values more than triple relative to wheat–maize and were significantly higher than the cereal-based spring maize→wheat–maize and ryegrass-sweet sorghum→wheat–maize rotations (0.25–0.29). Significant relationships occurred among the key indicators across the rotation systems (Fig. 6c). Soil health was positively corelated with crop yield ($r = 0.79$), economic benefits ($r = 0.85$), and nutrition score ($r = 0.90$), but was negatively correlated with net GHG emissions ($r = -0.83$). Soil health was also positively correlated with soil carbon sequestration ($r = 0.79$) and soil biodiversity ($r = 0.67$), contributing substantially to the trade-off against net GHG emissions ($r = -0.83$).

## Discussion

The empirical evidence from the 6-year field experiment demonstrated significant positive impacts of diversified crop rotations on various agroecosystem functions and services. Based on the study's findings and double-cropping planted area[25], we estimated that diversified cropping systems could reduce net $CO_2$-eq emissions by $106.8 \pm 31.7$ million tonnes annually, offsetting 5.6% of the annual GHG emissions associated with China's food system (1.9 billion tonnes in 2020 reported by FAO[26]). The primary reason behind this reduction is the decrease in synthetic N fertilizer use by 3.6 million tonnes. Farmers in the North China Plain region could benefit from 20% increases in annual net income, equivalent to 84 billion Yuan ($11.6 US billion) in total. Moreover, these diversified rotations will increase Cornell Soil Health Assessment scores by 45%, due to increased C sequestration rates compared to the winter wheat–summer maize rotation. Furthermore, wheat and maize yields may increase by 32%, equivalent to 73.5 million tonnes per year if planted following alternative crops such as sweet potato, soybean or peanuts; this would make about 36.1 million tonnes of additional straw biomass available annually for alternative uses, such as feed, bioenergy, or enhancing soil carbon stocks.

The three-dimensional benefits (grain and protein yields, GHG emissions, and soil health) of crop diversification in the winter wheat–summer maize rotation system on the North China Plain will add significant 'social novelty' to the challenge of bringing essential food nutrients to dining tables without adversely affecting soil health. The social novelty is far beyond the agricultural benefits of crop diversification that are broadly recognized worldwide. For example, long-term studies in North America show that rotational diversification increases crop yields, reduces the impact of adverse weather on ecosystem productivity[27], and enhances system robustness[28]. Studies in Europe and Africa demonstrate that field-scale diversity has a substitutive interaction with fertilization leading to increased yields at low N fertilizer doses[29–32]. Several Chinese studies implementing crop diversification in various forms or scales show that farmers can adopt multiple strategies to increase farming resilience and economic

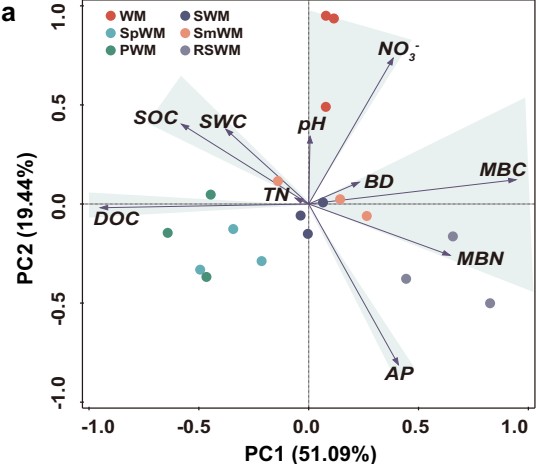

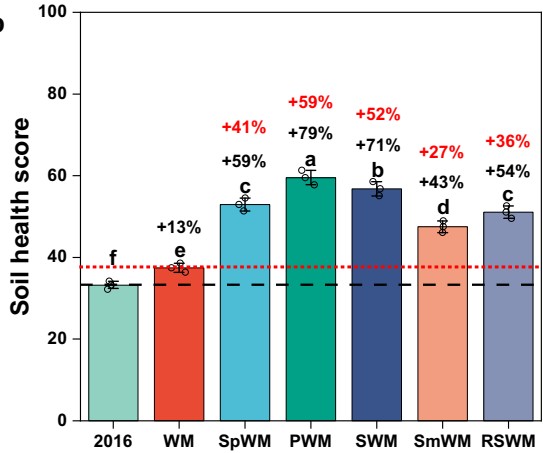

**Fig. 4 | Soil health assessment for different rotation systems. a** Principal component analysis (PCA) for eigenvalues of the ten key soil properties, including soil organic carbon (SOC), dissolved organic carbon (DOC), total nitrogen (TN), $NO_3^-$-N, available P (AP), pH, microbial biomass N (MBN), microbial biomass carbon (MBC), bulk density (BD), and soil water content (SWC). Each soil property was normalized as an individual CSHA (Cornell Soil Health Assessment) contributor to the overall soil health score. **b** CSHA score for each rotation. In **b**, the whiskers in each column represent the standard deviation of three replications; circles are the corresponding data points for the bar chart, $n = 3$. One-way ANOVA with two-sided and post-hoc test was conducted for the significance test among different treatments.

Different lowercase letters denote significantly different groups of crop rotations at $P < 0.05$. The red dashed line with red percentages is the baseline of the WM rotation. The black dashed line with black percentages is the baseline of 2016. Treatment abbreviations: WM winter wheat–summer maize (control case), SpWM sweet potato→winter wheat–summer maize rotation, PWM peanut→winter wheat–summer maize rotation, SWM soybean→winter wheat–summer maize rotation, SmWM spring maize→winter wheat–summer maize rotation, RSWM ryegrass-sorghum→winter wheat–summer maize rotation. Source data are provided as a Source Data file.

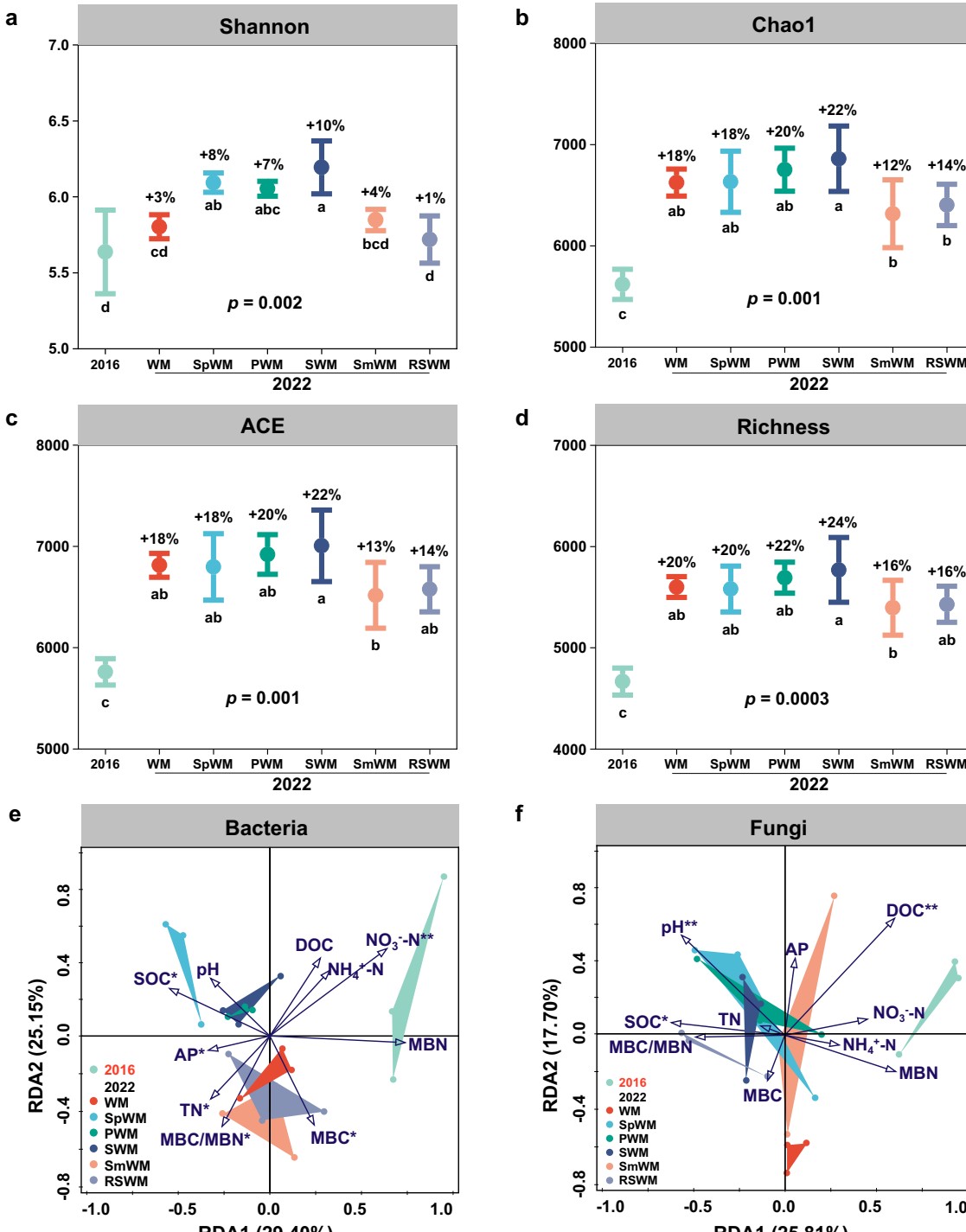

**Fig. 5 | Soil microbe community diversity and soil properties in response to rotation. a** Shannon Index; **b** Chao 1; **c** ACE index; **d** OTU richness measured in the six crop rotations from 2016 to 2022; **e** Redundancy analysis (RDA) of bacterial OTU and soil properties; **f** RDA of fungal OTU and soil properties. The percentage values in **a**–**d** represent the increment from 2016 to 2022; dot is the mean value, whisper line bars on each dot are the standard deviations for three replications. In **a**–**d**, one-way ANOVA with two-sided and post-hoc test was conducted to determine significant differences among treatments. Different lowercase letters below pillars indicate significant differences between rotations at $P < 0.05$. Charts **e** and **f** show redundancy analysis with colored dots and triangles indicating rotation, vectors indicating soil properties including soil organic carbon (SOC), dissolved organic carbon (DOC), total nitrogen (TN), $NO_3^-$-N, $NH_4^+$-N available P (AP), pH, microbial biomass carbon (MBC), microbial biomass N (MBN), and the MBC to MBN ratio. Treatment abbreviations: WM winter wheat–summer maize (control case), SpWM sweet potato→winter wheat–summer maize rotation, PWM peanut→winter wheat–summer maize rotation, SWM soybean→winter wheat–summer maize rotation, SmWM spring maize→winter wheat–summer maize rotation, RSWM ryegrass–sorghum→winter wheat–summer maize rotation. Source data are provided as a Source Data file.

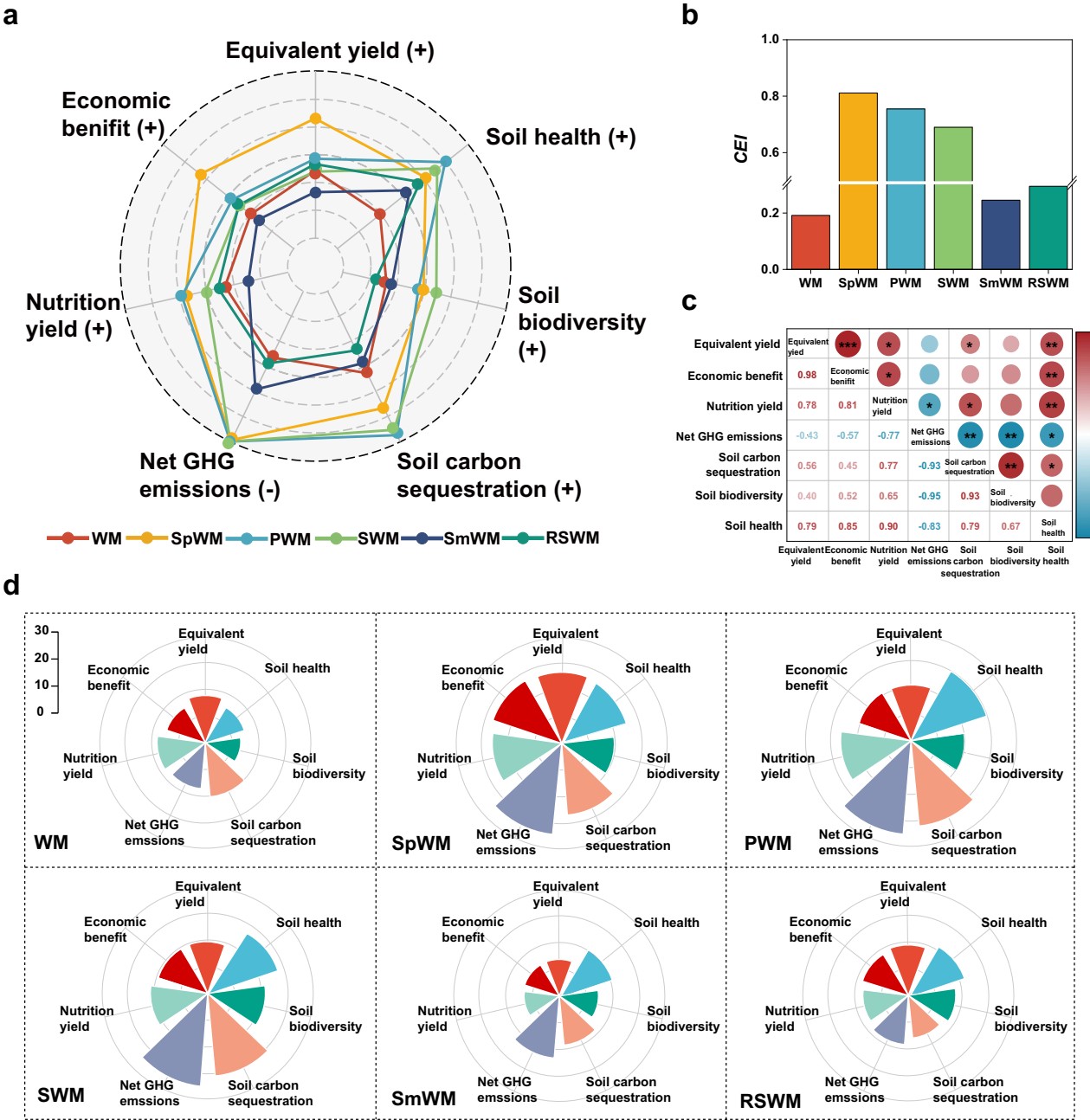

**Fig. 6 | Comprehensive assessments of the different rotation systems. a** radar map of multiple objective analysis to assess the various functions of the rotations; **b** comprehensive evaluation index (*CEI*, Eqs. 15−24 in Method, supplementary material); **c** synergy and trade-off relationship among the key variables across the different crop rotations; **d**, Nightingale Rose Charts of multiple objectives analysis to assess the detailed functions 1for each crop rotation. In **a**, values are normalized. (+), positive variables, (−), negative variables. Equivalent yield re-expresses crop yield as equivalent to the grain yield of wheat; Economic benefit denotes the crop yield multiplied by its market price minus consumable cost; Nutrition yield includes 17 nutritional components of each crop such as crude protein, fat content, macro- and micro-elements, and vitamins[68]. Soil carbon sequestration denotes changes in soil carbon stock in the 0–90 cm layer from October 2016 to October 2022; Net GHG emissions denotes the net GHG budgets including direct and indirect GHG emissions minus soil carbon sequestration; Soil biodiversity is represented by the Shannon Index of bacterial and fungal communities; Soil health score is calculated from 10 indicators using the overall Cornell Soil Health Assessment (CSHA) matrix combined with PCA[81–83]. Treatment abbreviations: WM winter wheat–summer maize (control case), SpWM sweet potato→winter wheat–summer maize rotation, PWM peanut→winter wheat–summer maize rotation, SWM soybean→winter wheat–summer maize rotation, SmWM spring maize→winter wheat–summer maize rotation, RSWM ryegrass–sorghum→winter wheat–summer maize rotation. Source data are provided as a Source Data file.

benefits[28,33]. Overall, cropping diversification offers a comprehensive systems' approach to enhance agroecosystem productivity and adaptability to changing climates worldwide.

A major factor in designing diversified cropping systems is to ensure an increase in soil C sequestration—as this plays a crucial role for soi health, in climate change mitigation and achieving C neutrality[34,35]. We found that the diversified rotations with sweet potato, peanut and soybean significantly increased soil C after 6 years of rotation nourishment compared to the traditional winter wheat–summer maize double-cropping system. Introducing sweet potatoes and legumes into these rotations has been shown to stimulate microbial growth, increased C use efficiency[36], and promoted the formation of highly stable mineral-associated soil C complexes[37]. Labile litter compounds from these rotational crops are the dominant

source of microbial products that affect stable SOC with a high capacity for C stabilization[38]. Furthermore, our findings highlight a close correlation between soil health and nutrition yield, underscoring the interconnectedness of both dimensions and their reliance on agricultural practices[39].

A noteworthy aspect of the study is that we measured SOC concentrations to 90 cm depth. Researchers often focus on topsoil layers (0–20 cm) when studying SOC status, potentially masking some treatment effects, especially in crops with different root masses at various soil depths. We found that SOC in the 0–20 cm soil layer accounted for 51% of the total SOC, 0–30 cm layer accounted for 67%, and 0–50 cm layer accounted for 80%. Our results highlight the importance of subsoil C stocks when assessing soil C storage for a more precise estimation of the contributions of soil C sequestration in mitigating climate change.

Lowering GHG emissions without reducing crop yields is a major challenge for global agriculture. We found that the legume-based (peanut→wheat–maize, soybean→wheat–maize) rotations significantly reduced $N_2O$ emissions by 30–42% compared to the conventional cereal-based wheat–maize rotation while significantly increasing system productivity (grain and protein yields). Legume crops improve soil N cycling[40], but may also increase $N_2O$ emissions, specifically following residue incorporation[41,42]. However, legume cultivation requires significantly less fertilizer than cereal crops such as maize or wheat. Therefore, the legume-based rotations with peanut and soybean required 37% less fertilizer than conventional wheat–maize, explaining the overall reduction in $N_2O$ emissions for the legume-based rotations as N fertilizers were the main source of $N_2O$ emissions in our study[40]. The manufacture of inorganic N fertilizer remained the most significant contributor to total GHG emissions (about 36–39%). Similarly, we found that the diversified rotations with sweet potato, soybean, and peanut significantly reduced net GHG emissions by 75–92% compared to the wheat–maize, spring maize→wheat–maize, and ryegrass-sweet sorghum→wheat–maize systems. In a study in Western Canada, cereals rotated with legumes decreased net GHG emissions by 17–35% compared to cereal monocultures[43] due to improved N uptake associated with organic N mineralization released from legume residues[44]. In Europe, increasing legume production by introducing legumes into cereal rotations effectively improved plant-based protein while reducing environmental impacts[45,46]. In the US corn belt, diversified maize and soybean systems maintained or increased maize yields while reducing N losses to the environment[47].

Introducing shallow-rooted crops like sweet potato and annual legumes in rotation with deep-rooted crops such as winter wheat enables the use of soil nutrients across the entire rooting zone, improving nutrient use efficiency and reducing soil nutrient losses through leaching. Alternating deep- and shallow-rooted crops also enhances root distribution, increases soil porosity and permeability, reduces soil bulk density[48], and improves soil aggregate stability[49] to anchor SOC[50]. Furthermore, diversified crop mixtures alter the quality and quantity of organic matter entering the soil as plant litter, enhancing the stabilization of microbial-derived compounds that provide positive feedback for plant growth. The distinct microbial communities associated with diverse crop mixtures can promote soil C cycling[51]. In our study, diversified legume–cereal and sweet potato–cereal rotations significantly increased the α-diversity of soil bacterial and fungal communities after 6 years of rotation. The increased fungal diversity resulted from a higher number of species, while the increased Shannon's diversity index in the bacterial community resulted from changes in the relevant abundance of certain species. Diverse crops stimulate bacterial community activities in the rhizosphere[52], as the residue root exudates and plant litter from rotational crops provide a greater diversity of residue carbon substrates, supporting the growth of diverse soil microorganisms[53]. We found that the rotations with sweet potato, peanut, and soybean significantly increased fungal community diversity compared to winter wheat–summer maize, shaping soil nutrient profiles with different carbon substrates that impacted microbial diversity. Moreover, diversified crop rotations potentially reduce soil agrochemical contamination due to decreased application of synthetic N fertilizer and herbicides, favoring soil microenvironments[54,55]. Other studies have demonstrated the positive influence of diversified rotations on soil health. Maize yields and SOC content in diversified rotations were more resilient than those found in monoculture[56], potentially increasing adaptability to climate change[57]. Diversified agroecosystems are considered an economically viable alternative to business-as-usual maize–soybean rotations in the Midwest US[58], with cover crops, perennials, and small grain cereals enhancing soil health by 32–49% and crop productivity by 16–29%[59].

The novelty of this study is its highly diversified crop rotations, with cash crops and legumes replacing cereal monocultures, together with rotating shallow-rooted crops with deep-rooted crops. It positively affects ecosystem services and functions, partly compensating for the lower services in less diversified cereal-based rotations. Crop diversification offers significant benefits in reducing GHG emissions and has a synergistic effect on plant biomass and protein production, soil health, and microbial community biodiversity. Diversification reduced crop inputs, particularly N fertilizer and irrigation amounts, minimizing environmental costs while increasing farmers' incomes. Thus, diversified crop rotations can serve as an effective strategy for sustainable food production in areas worldwide with environments similar to the North China Plain.

The scientific evidence presented in this study holds significant potential for informing and reinforcing current policies and incentive schemes to improve the green transition of farming systems in China. Specifically, China set objectives to promote soybean production (initiated in 2019), implement an action plan for zero increase in fertilizer and pesticide use (launched in 2015), and develop strategies to reduce agricultural GHG emissions with the ultimate goal of transforming agricultural systems into zero 'emitters' by 2060. When designing effective diversified cropping systems, it is important to consider local environmental conditions to balance land use for food production with the other functions of an ecosystem. Optimizing crop configuration is essential—through social, economic, and environmental dimensional planning within a complex framework of various indicators to ensure long-term sustainability. We recommend that developing and adopting diversified cropping systems should be a key consideration in agricultural policy setting and a top priority for on-farm decision-making, as they are critical for achieving long-term food production resilience, maintaining soil health, and ensuring environmental sustainability. Given that climate variability will increase, multi-year assessments of the sustainability of agricultural production systems remain tentative, and funding should be sought for additional observation sites for long-term monitoring.

## Methods

The field experiment was established in October 2016 at the Luancheng Agro-Ecosystem Station of the Chinese Academy of Sciences (37°50′ N, 114°40′ E; altitude, 50.1 m) in Luancheng County, Hebei Province (Fig. S5a), which represents the agricultural production and climate conditions of the North China Plain. The experimental area experiences a warm, temperate zone, semi-humid, monsoon climate with a frost-free period of 200 days. The annual mean air temperature was $14.7 \pm 1.0\,°C$ over the last 20 years, and the annual average precipitation was $472 \pm 161$ mm over the last 62 years (Fig. S5b). Fig. S5c presents the average monthly precipitation and air temperatures from 2016 to 2020. The experimental site had loam soil with sandy loam in the surface layers, light/medium loam at 40–80 cm depth, and light clay below 80 cm. The 0–10 cm soil layer before the experiment start has pH $7.6 \pm 0.2$, $240 \pm 42\,\mu S\,cm^{-1}$ EC, $1.49 \pm 0.07\,g\,cm^{-3}$ bulk density,

$0.42 \pm 0.03$ cm$^3$ cm$^{-3}$ field capacity, $0.10 \pm 0.02$ cm$^3$ cm$^{-3}$ wilting point, $1.06 \pm 0.14$ g kg$^{-1}$ total N, $11.5 \pm 0.4$ g kg$^{-1}$ SOC, $9.3 \pm 3.1$ mg kg$^{-1}$ available phosphorous, and $109.6 \pm 24.1$ mg kg$^{-1}$ available potassium. Prior to the experiment, the core food crops on the same fields were wheat and maize grown in a 12-month double-cropping system that is popular regionally. From the start of the experiment, the sites were managed with optimized irrigation and seeding techniques, with all residues returned to the soil during the winter wheat–summer maize year.

The experiment comprised eight crops: cereals winter wheat (cv. Kenong 2009), summer maize (cv. Jundan 20), and spring maize (cv. Jundan 20); legumes soybean (cv. Shidou 12) and peanut (cv. Jihua 4); cash/forage: sweet potato (cv. Shangshu 19), ryegrass (cv. Dongmu 70) and sorghum (cv. Jintianza 3). There were five newly designed diversified crop rotations: [sweet potato→winter wheat–summer maize rotation (SpWM, 2-year cycle); peanut→winter wheat–summer maize rotation (PWM, 2-year cycle); soybean→winter wheat–summer maize rotation (SWM, 2-year cycle); ryegrass-sweet sorghum→winter wheat–summer maize rotation (RSWM, 2-year cycle); spring maize→winter wheat–summer maize rotation (SpWM, 2-year cycle)]. The regionally dominant winter wheat–summer maize double-cropping system (WM, 1-year cycle) was the control. '→' denotes crop rotation across years and '–' denotes crop rotation within a year. Unlike pure winter wheat–summer maize, the alternative rotations included fallow seasons for parts of their non-winter wheat–summer maize years (Fig. S6).

Each rotation was replicated three times, resulting in 18 plots established using a randomized complete block design (Fig. S6). All cycles were repeated on their respective plots from October 2016 to October 2022. The winter wheat–summer maize double-cropping control rotation completed six cycles, while the other five rotations completed three rotation cycles. Supplementary Table S1 lists the total amounts of inputs, including N, P, K, irrigation, diesel for tillage, electricity for irrigation, pesticide, seeds, and labor for each crop in each rotation. Supplementary Data 1 details fertilizer and irrigation amounts and timings for each crop in each rotation in each year. Basal N applications (as urea) were broadcast and plowed (20 cm depth) into the soil before seeding. Topdressing of N was performed at the key growth stages with specific amounts summarized in Supplementary Data 1.

## Equivalent yield and economic benefit

Crops were harvested at maturity with a conventional combine harvester with an automated weighing system, which measured the total crop yield of each plot. After air-drying to a constant moisture content (13%), wheat and maize grain yields were recorded[60,61]. Sorghum biomass yield (fresh weight) per plot was weighed to represent silage moisture of 30% (70% dry matter)[61]. Ryegrass dry matter was determined in $1 \times 1$ m sub-sample after drying at 85 °C until constant weight[62]. Sweet potato yield (fresh weight) was obtained by weighing the fresh root tubers harvested from each plot using a single row harvester[63]. All mature pods in each plot were collected to determine peanut yield accurately. Ten uniform peanut plants were used for investigating the pod numbers per plant. All harvested peanut pods were air-dried to a water content of approximately 14% and weighed to calculate the pod yield[64]. After harvesting each plot, soybean seeds were cleaned and weighed, with the grain yields adjusted to 13.0% moisture content[65].

For yield comparisons among different crop products, the equivalent yields of the six crop rotations were calculated by multiplying the crop yield by the market price relative to that of winter wheat in the same year[66,67]. Each cropping system's equivalent yield, economic benefit, and protein yield were determined as follows:

$$EY_{ij} = Yield_{ij} \times \frac{Price_{ij}}{Wheat\ price_j} \quad (1)$$

$$EB_{ij} = Yield_{ij} \times Price_{ij} \times \frac{CPI_j}{CPI_{2008}} - Cos\,t_{ij} \quad (2)$$

$$PC_i = Yield_i \times \beta_i \quad (3)$$

where $EY_{ij}$ is equivalent yield of crop $i$ at year $j$ (kg ha$^{-1}$), wheat price$_j$ is price of winter wheat in year $j$ adjusted for China National Consumer Price Index (CPI, relative Yuan), economic benefit (EB) is within-year crop yield multiplied by its corresponding price in Chinese Yuan minus crop production costs (Yuan ha$^{-1}$), $j$ is a given year, $PC_i$ is total protein content of crop $i$ (kg ha$^{-1}$), and $\beta_i$ is protein concentration per kg crop product (%) referred from previous studies[67,68]. (For comparison, the following exchange rate was used: \$1 US = 6.95 Chinese Yuan and 1 € = 7.60 Yuan in May 2023.) Each indicator for one crop rotation was summed across the crops within one crop rotation cycle. The CPI was used to adjust crop prices across years to eliminate inflation or deflation as a source of variability between cases in different years. Table S2 lists the crop prices and CPI values for individual years and crops, and Table S3 lists each crop's protein content.

## Gas flux and net GHG emissions

Soil GHG (N$_2$O and CH$_4$) fluxes were measured using the static chamber method[69]. Plastic frames with water-sealing grooves were inserted 5 cm into the soil between crop rows, and chambers ($20 \times 20 \times 30$ cm) were placed on top of the frames during measurements. For flux measurements, gas samples (>30 mL per sample) were taken between 9:00 and 10:00 am at specific time intervals (0, 12, 24, and 36 min after chamber closure) using 100 mL plastic syringes attached to a three-way stopcock. Gas samples were collected about once every 7 days from sowing to harvest, with more frequent measurements (every 3 days) after N fertilizer application and precipitation events. The headspace air temperature in the chamber was recorded with a thermometer. Gas samples were analyzed using a gas chromatograph (Agilent 7890 A, Agilent Inc., USA) equipped with a flame-ionization detector for measuring CH$_4$ at 200 °C and an electron capture detector for measuring N$_2$O at 330 °C. The column temperature was set to 55 °C. CH$_4$ was separated with a Porapak Q column and detected by the flame-ionization detector. N$_2$O was detected by a micro electron capture detector (μECD). The carrier gas was high-purity N$_2$ (99.999%) with a flow velocity of 20 mL min$^{-1}$. A purge gas (5% CO$_2$ in N$_2$) was used to avoid interference with CO$_2$. The gas flux was calculated using the linear approach. Soil N$_2$O and CH$_4$ fluxes for each rotation across the 6 years are illustrated in Figs S1 and S2.

For the net GHG emissions (kg CO$_2$-eq ha$^{-1}$) estimation, we considered indirect GHG emissions of crop inputs from a life cycle assessment, direct N$_2$O and CH$_4$ emissions from the soil, and changes in soil carbon stock from 0–90 cm during the 6-year experiment. The system boundary in the life cycle assessment was set from manufacture, storage, and delivery of external inputs such as fertilizer to crop harvest, including soil carbon storage change (Fig. S7). Potential emissions related to the logistics of transporting, exporting, or marketing grain products beyond the farm gate were excluded as they were considered outside the system boundary.

Indirect GHG emissions (CE) result from manufacturing, storing, transporting, and delivering agricultural inputs, such as fertilizer (N, P, K), pesticides, diesel, and electricity to the farm gate, as follows:

$$CE = \sum_{k=1}^{n} (D_k \times C_k) \quad (4)$$

where CE (kg CO$_2$-eq ha$^{-1}$ yr$^{-1}$) is the indirect GHG emissions for one crop within a given rotation, $D_k$ is the amount of the $k^{th}$ input (Table S1), and $C_k$ is the corresponding CO$_2$-eq emission coefficients (see Table S4 for the manufacture, storage, and delivery of agricultural inputs).

Direct $N_2O$ and $CH_4$ fluxes from soil were calculated from October 2016 to October 2022, as follows:

$$f = \frac{M}{V_0} \frac{T_0}{T} \frac{P}{P_0} H \frac{d_c}{d_t} \times 60 \qquad (5)$$

where $f$ (ug m$^{-2}$ h$^{-1}$) is GHG emission fluxes of $N_2O$ and $CH_4$ emissions, $M$ is the molecular mass of the specified gas (44 g mol$^{-1}$ for $N_2O$ and 16 g mol$^{-1}$ for $CH_4$), $V_O$ (m$^3$mol$^{-1}$) is the molar volume of an ideal gas, $T_O$ (K) is the ideal gas temperature, $T$ (K) and $P$ (kPa) are the air temperature and pressure in the chamber at the sampling time, $P_O$ (kPa) is the standard air pressure, $H$ (m) is the height of chamber, $d_c/d_t$ (ppm min$^{-1}$) is the slope of the linear regression curve for the change of headspace gas concentrations during times of chamber closure, and 60 is min h$^{-1}$. Total GHG emissions during each crop growth and fallow period were calculated using linear interpolation[70].

Cumulative $N_2O$ and $CH_4$ emissions during each crop's growing season or fallow period were calculated as:

$$F = \sum_{i=1}^{n} \left[ \frac{(f_{i+1} + f_i)}{2} \times (t_{i+1} - t_i) \right] \times 24 \qquad (6)$$

where $F$ is the cumulative emission of one of the two gases from one experimental plot (kg ha$^{-1}$) over a specified period, $f$ is the gas emission flux (kg ha$^{-1}$ h$^{-1}$), $t$ is the sampling time in days, $i$ is the $i^{th}$ sampling from the plot, and 24 is the conversion coefficient between hours and days (h day$^{-1}$).

The global warming potential (GWP) of seasonal soil $N_2O$ and $CH_4$ emissions, combining Eq. (7) results and re-expressed in $CO_2$-eq[71], was calculated as:

$$GWP_{N_2O + CH_4} = N_2O \times 273 + CH_4 \times 27 \qquad (7)$$

The annual change in SOC stock ($\Delta C$; kg $CO_2$-eq ha$^{-1}$ yr$^{-1}$) from October 2016 to October 2022 was estimated for each crop rotation, as follows:

$$\Delta C = \frac{(T_1 - T_2) \times \frac{44}{12}}{n} \qquad (8)$$

where $T_1$ and $T_2$ (t ha$^{-1}$) are the total soil carbon storage values in the 0–90 cm soil layer in October 2016 and October 2022, respectively, $n$ is the number of years of the study period, and 44/12 is the conversion coefficient from C into $CO_2$.

Net GHG emissions were equal to the sum of indirect ($CE$) and direct (GWP of $N_2O$ and $CH_4$) GHG emissions minus soil carbon sequestration ($\Delta C$) when soil carbon was the sink, calculated as follows:

$$Net\ GHGs = CE + GWP_{N_2O + CH_4} - \Delta C \qquad (9)$$

## Soil physiochemical properties, microbial diversity and soil health scoring

Soil samples (0–20 cm depth) were collected at the beginning of the experiment (October 2016) and at the end of the summer maize harvest in October 2022 to measure soil physicochemical properties: including soil pH, bulk density, soil water content, total nitrogen (TN), dissolved organic carbon (DOC), soil nitrate-N ($NO_3^-$-N), ammonium-N ($NH_4^+$-N), available phosphorus (AP), and microbial biomass carbon and nitrogen (MBC and MBN, respectively).

Soil pH was measured in a 1:5 (w/v) soil: water suspension as described by McLean (1982)[72]. TN was determined using the modified Kjeldahl method[73]. MBC and MBN were determined using the chloroform fumigation and extraction method[74]. DOC concentrations extracted from fumigated and unfumigated soil

samples were determined using a Multi 3100 N/C TOC analyzer (Analytik Jena, Germany). A modified Bremner's standard protocol was used to extract soil $NO_3^-$-N and $NH_4^+$-N using 2 M KCl[75], measured with a UV-1800 spectrophotometer (Mapada Instruments, Shanghai, China). Soil available P concentrations were determined using 0.5 mol L$^{-1}$ NaHCO$_3$ (pH 8.5) and the molybdenum-blue colorimetric method[76].

Soil organic carbon (SOC) concentrations in the 0–90 cm soil layer (0–10, 10–20, 20–30, 30–50, 50–70, and 70–90 cm increments) were measured after each crop harvest from October 2016 to October 2022, in samples collected by auger at five random locations per plot. Soil samples from each sampling depth within a plot were combined, dried at 105 °C for 24 h, sieved (2 mm), pretreated with 0.5 M HCl to remove carbonates[77], and ball-milled. SOC concentrations were determined using a vario Macro CNS Analyzer (Elementar, Germany). Soil bulk density in all soil layers down to 0.9 m in all plots was measured using the gravimetric method[78].

Soil samples were also collected before the start of the experiment (October 2016) and at the final summer maize harvest (October 2022) for DNA sequencing and microbial community composition and diversity measurements. Subsamples for DNA sequencing were immediately placed on dry ice and stored at −80 °C until DNA extraction, while the other subsamples were taken to the laboratory for chemical analysis. Soil DNA was extracted from 0.5 g freeze-dried soil using the Powersoil DNA Isolation Kit (Mo Bio Laboratories, Carlsbad, CA, USA) according to the manufacturer's protocol. DNA quality was assessed according to the 260/280 nm and 260/230 nm absorbance ratios using a NanoDrop ND2000 spectrophotometer (NanoDrop, ND2000, Thermo Scientific, 111 Wilmington, DE) and then stored at −80 °C for further molecular biology analysis.

Primer pairs−barcode-515F/806 R (5′-GTGCCAGCMGCCGCGGTAA-3′/5′-GCACTACHVGGGTWTCTAAT-3′)−were used to amplify the V3 + V4 region of the bacterial 16 S rRNA gene[79], yielding accurate taxonomic information with few biases among various bacterial taxa. Primers−ITS1 (5′-CTTGGTCATTTAGAGGAAGTAA-3′) and ITS2 (5′-TGCGTTCTTCATC-GATGC-3′)−were used to amplify the ITS1 region of the fungal rRNA gene. The PCR was carried out in a mixture (final volume, 50 μL) comprising 2 μL DNA template (1–10 ng), 5 μL of each 2 μM primer, 25 μL Premix Ex Taq (Takara Biotechnology), and 13 μL sterilized water. Thermal-cycling conditions were as follows: initial denaturation of 3 min at 94 °C, six touchdown cycles of 45 s at 94 °C, 60 s from 65 to 58 °C, 70 s at 72 °C, followed by 22 cycles of 45 s at 94 °C, 60 s at 58 °C, 60 s at 72 °C, with a final elongation of 72 °C for 10 min. Tag-encoded high-throughput sequencing of the 16 S and ITS genes from purified and quantified PCR products was performed by the Magigene Company (Beijing, China) using the Illumina MiSeq platform.

Raw sequences were quality processed using the Quantitative Insights into Microbial Ecology (QIIME) pipeline (version 2.1)[79]. Resampling operational taxonomic units (OTUs) was based on the minimum sequence numbers to correct the sampling effort before further analysis. The OTUs were clustered with a 97% similarity cutoff using the UPARSE pipeline (version 7.1)[80]. Taxonomic assignment was carried out using OTUs with SILVA (16 S) and Unite (ITS).

Alpha diversity was characterized by six indices (Shannon-Weaver, Chao1, ACE, Richness, Simpson, and Pielou), calculated using the OTU numbers for bacteria and fungi. (see Supplementary Information for detailed calculations).

The Cornell Soil Health Assessment (CSHA)[81–83] combined with principal component analysis (PCA)[84,85] was used to determine health scores for soils sampled in 2016 and 2022, which allowed comparing each plot's changes in soil health after 6 years of 'rotation nourishment'. The ten soil indicators (physical attributes: bulk density, soil water content; chemical attributes: soil pH, TN, SOC, DOC, $NO_3^-$-N, AP; biological attributes: MBC and MBN) were normalized as individual

CSHA scores[81]. The weights of individual CSHA scores were based on PCA of all soil indicators (Table S5), representing the sum of the eigenvectors derived from the first three principal components, which were selected based on the inflection point from a Scree plot and Kaiser's cut-off (eigenvalues > 1) (Fig. S8). These first three principal components accounted for 51.09%, 19.44%, and 15.04% of the data's total variation, with a cumulative variance of 85.57%, capturing most of the variation among the soil indicators. The soil health overall score (%) was computed as a weighted average of all individual CSHA scores, calculated as follows:

$$\text{Soil health score} = \frac{(A_1 \times w_1) + (A_2 \times w_2) + \cdots + (A_n \times w_n)}{w_1 + w_2 + \cdots + w_n} \quad (10)$$

where $A$ is the CSHA score for each individual soil indicator, and $w$ is the weighting factor for each soil indicator (Table S5).

### Entropy-TOPSIS for multiple function assessment of diversified crop rotations

The *CEI* was determined using crop yield, nutritional yield (detailed in Eqs. 1–5 in SI Methods), economic benefit, soil health, C sequestration, microbial biodiversity, and net GHG emissions based on Entropy-TOPSIS (Technique for Order Preference by Similarity to an Ideal Solution).

Entropy-TOPSIS is a multi-objective decision analysis tool for identifying the most feasible solution from a set of options. It defines the most positive and negative ideal solutions of a problem, with the most feasible solution representing the solution that is the most positive and furthest from the negative ideal solution[86–88]. The entropy weight method was selected as an objective determination of the index weight[87,89] to overcome the influence of subjective factors caused by the Delphi method[90] or the analytic hierarchy process when determining the weight of the evaluation index of the traditional TOPSIS method. The index weight is then used to calculate the comprehensive evaluation rank of each crop rotation using TOPSIS:

(1)  Normalize the indicators:

$$b_{ij} = \frac{x_{ij} - x_{\min}}{x_{\max} - x_{\min}} \, (\text{positive indicator}) \quad (11)$$

$$b_{ij} = \frac{x_{\max} - x_{ij}}{x_{\max} - x_{\min}} \, (\text{negative indicator}) \quad (12)$$

where $b_{ij}$ is the normalized value of the $j$th index in the $i$th crop rotation (positive indicators include equivalent yield, economic benefit, nutrition yield, soil biodiversity including bacteria and fungi, soil health score, and soil carbon sequestration; negative indicator is net GHG emissions), $x_{ij}$ is the average value from three measured replications of the $j$th index in the $i$th crop rotation, and $x_{max}$ and $x_{min}$ are the maximum and the minimum values of a single index, respectively.

(2)  Calculate the entropy value ($H_i$) of each evaluation index:

$$H_i = \frac{-1}{\ln m} \sum_{j=1}^{m} f_{ij} \ln f_{ij} \quad (13)$$

$$f_{ij} = \frac{b_{ij}}{\sum_{j=1}^{m} b_{ij}} \quad (14)$$

where $i = 1, 2, \ldots, n$ and $j = 1, 2, \ldots, m$. When $b_{ij} = 0$, $\ln f_{ij}$ is modified as follows:

$$f_{ij} = (1 + b_{ij}) / \sum_{j=1}^{m} (1 + b_{ij}) \quad (15)$$

(3)  Calculate the entropy weight ($W_j$) of the evaluation index:

$$W_j = \frac{1 - H_j}{n - \sum_{i=1}^{n} H_i} \quad (16)$$

(4)  Construct a weighted normalized decision matrix ($Z$) using the normalized matrix $f_{ij}$ and the weights of each index $W_j$:

$$Z = (W_j \times f_{ij})_{n \times m} = \begin{bmatrix} z_{11} & z_{12} & \cdots & z_{1m} \\ z_{21} & z_{22} & \cdots & z_{2m} \\ \cdots & \cdots & \cdots & \cdots \\ z_{n1} & z_{n2} & \cdots & z_{nm} \end{bmatrix} \quad (17)$$

(5)  Determine the positive ideal solution vector $Z^+$ and the negative ideal solution vector $Z^-$ using the positive ideal solution ($Z_j^+$) and the negative ideal solution ($Z_j^-$):

$$\begin{aligned} z_j^+ &= \max(z_{1j}, z_{2j}, \ldots, z_{nj}) \\ z_j^- &= \min(z_{1j}, z_{2j}, \ldots, z_{nj}) \end{aligned} \quad (18)$$

(6)  Calculate the Euclidean distance between each index and the positive and negative ideal solutions.

$$\begin{aligned} D_i^+ &= \sqrt{\sum_{j=1}^{m} (z_j^+ - z_{ij})^2} \\ D_i^- &= \sqrt{\sum_{j=1}^{m} (z_j^- - z_{ij})^2} \end{aligned} \quad (19)$$

(7)  Calculate the comprehensive evaluation index $CEI_i$ of each crop rotation. The proximity between the evaluation object and the optimal scheme is calculated as follows:

$$CEI_i = \frac{D_i^-}{D_i^+ + D_i^-} \, (0 \leq CEI_i \leq 1) \quad (20)$$

### Statistical analysis

Statistical analyses were conducted in SPSS version 21 software (IBM's Statistical Product and Service Solutions) and R version 4.3.1[91]. The Kolmogorov-Smirnov test was used for testing normal distribution, followed by an analysis of variance (ANOVA) to determine the statistical significance of differences between mean values. Post-hoc comparisons were conducted using the least significant difference (LSD) test at the 5% probability level. The correlation matrix of indicators for each rotation was calculated with Pearson's classic method and visualized using the Performance Analytics package in R 4.3.1 and R Studio 1.8.0. Principal component analysis (PCA) was used to calculate the eigenvalues and classify each soil indicator in each rotation using the vegan package, visualized using the ggplot2 package in R 4.3.1. The α-diversity indexes of the soil microbial community were calculated using the vegan and picante packages, visualized using R 4.3.1. A radar map illustrating the performance of the multiple objective analysis was produced to assess the various functions of each rotation, visualized using ggradar packages. A Nightingale Rose Chart was used to assess the functions of each crop rotation, visualized using ggplot2 package.

The results presented in the paper were derived predominantly from a 6-year field experiment undertaken at the Luancheng research station, a typical representative site for production systems and environmental characteristics in the North China Plain. Multi-location experiments will ensure the spatiotemporal effects of diversified crop rotations on system productivity, environmental impacts, and socio-economic performance. Moreover, inter-annual weather conditions are highly variable, and this may intensify in the future. Thus, longer term monitoring at multiple sites would be valuable for reducing the

uncertainty associated with extreme weather events and climate change.

## Reporting summary
Further information on research design is available in the Nature Portfolio Reporting Summary linked to this article.

## Data availability
The raw reads from Illumina sequencing described in this study, are available at NCBI under the accession no. SRR26083954-26083974 (16 S rRNA) and SRR26083997-26084017 (ITS). The dataset generated in this study has been deposited in the database under accession code on Figshare at https://doi.org/10.6084/m9.figshare.24793563. Source data are provided with this paper.

## Code availability
Code scripts used in this study are available at: https://figshare.com/s/5c07d2ac7c8dc4a5b678.

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

## Acknowledgements

We thank J.Z., N.Q., P.H., F.L., Y.C. and R.C. from China Agricultural University for field work contributions, and Dr. Y.S. for assisting the analysis of soil microbial diversity. This work was financed jointly by the National Natural Science Foundation of China (Grant No.32071975 to X.Y., 52239002 to T.D. and 41830751 to X.J.), the Hebei Province Key Research and Development Program of China (20326411D-1 to X.Y.) and Pioneer Center for Research in Sustainable Agricultural Futures (Land-CRAFT) (DNRF grant number P2 to K.B.B.).

## Author contributions

X.Y., S.K., T.D., and X.J. designed the study. X.Y. designed the field experiment. X.Y., J.X., S.L., L.X. and Y.S. facilitated data collection. X.Y., J.X., X.J. and K.B.B. do the data analysis. X.Y., X.J., K.B.B., Y.G. and K.H.M.S. wrote the manuscript. Y.G., K.H.M.S., S.P. and T.S.S. improved and revised the writings. All authors discussed the results and contributed to the final manuscript.

## Competing interests

The authors declare no competing interests.
