## [Peer Review File · Nature Communications]

Reviewers' Comments:

Reviewer #1:

Remarks to the Author:

What are the noteworthy results?

As a reviewer, I find that the results of this six-year field experiment are quite interesting. The most noteworthy results are: (1) more diversified food/crop production systems increased farms' income while not reducing crop yields as compared with less diversified or traditional cropping systems; (2) diversification significantly reduced greenhouse gas emissions due to the inclusion of legume crops in crop rotation reducing the use of N fertilizers; (3) cropping system diversification enhanced soil health due to improved soil microbial activities. These results are novel and informative!

Will the work be of significance to the field and related fields?

Yes, the work is indeed of significance to the related field. I would think that the work, once published, will have a significant impact on policy-making regarding crop diversification and emission mitigation and improvement of soil health.

How does it compare to the established literature? If the work is not original, please provide relevant references

This work is original. Compared with the existing literature in which numerous publications on the similar topic describing the effect of crop and cropping systems diversification on crop yield, pest control, and systems resilience etc, but little information has been documented in regard to how crop diversification can reduce carbon emissions and at the same time enhancing soil health. The latter is a hot topic currently in the world research community.

Does the work support the conclusions and claims, or is additional evidence needed? Are there any flaws in the data analysis, interpretation and conclusions? - Do these prohibit publication or require revision? Is the methodology sound? Does the work meet the expected standards in your field? Is there enough detail provided in the methods for the work to be reproduced?

This paper, as it is in its current form, is not ready to publish yet because I find a number of errors, unclear statements, and I would suggest authors address them before it is accepted for publication even although they are mostly minor concerns:

1) Line 35, the statement "improved the system's greenhouse gas balance by 88%". Please explain what this meant? Did you really mean improved GHG 'emission balance' by 88%? i.e., CO₂ eq emissions: carbon sequestration?

2) Line 36, "legume inclusion" not clear to me

3) Line 40, "while benefiting the environment", please more specific, what environmental benefits they offered?

4) Line 58, "double-cropping systems" are not always bad, they can be beneficial in many other regions. Please modify the statement.

5) Line 65, change "agricultural commodities" to "agricultural products".

6) Line 68, add a hyphen between legume and diversified

7) Lines 72-73: please add reference(s) about the statement "the North China Plain (NCP) —the food basket of China—one of the most intensively cultivated regions in the world"

8) Lines 67, 78, 81, 722, you have used the following inconsistent terms on: farmer income, farmers income, farms' incomes, or farm incomes! Please be consistent!

9) Line 87, what did 'environmental cost' mean?

10) Line 88, change "we concluded that" into "our results demonstrate that

11) Fig. 1 caption, please change "diversified rotation systems with cash and legume crops..." into "rotation systems diversified with cash and legume crops..."

12) Fig. 2 caption, line 729-730: please change to: "the dash lines in d-f are the baseline values of the WM rotation".

13) Fig. 3 caption, line 739, please change to "e, changes in soil carbon stock in the 0–90 cm depth..."

14) Fig. 3, panels d and f, please switch X-axis with Y-axis to make them consistent with panel e. in fact, you can not switch the two axes between the "cause" and the "effect".

15) Fig. 3 panel, not clear to me why it was called "net GHG emissions" as it presents both direct and indirect emissions and soil carbon sequestration. Would be "net" an end product?

16) Line 742, delete the words "on each indicator"

17) Line 753, please delete the word "respectively."

18) Fig 4, panel b, please add a baseline of the WM rotation, like what were done in Fig. 2d-f. Also,

- please add percentage values in b to indicate the increment from
- 19) 2016 to 2022, like what were done in Fig. 5, panels a-d.
 - 20) Line 760, please correct the statement into: Treatment abbreviations are the same as those in Fig. 2.
 - 21) Fig. 6, panel b. The CEI index calculation using eqs. 15-24 shown in the SI file, where x_{ij} is the average value of the j th index in the i th crop rotation. Not clear to me, average from what? From replicates? If so, please add $n=3$ in the statement, if not, please tell readers how it was averaged.
 - 22) The calculation of CEI, as described in lines 131-170 of the SI file, please provide more details about how the CEI was processed and using what software program?
 - 23) Fig. 6, panel c, stated that "Nutrition yield includes 17 nutritional components of each crop such as crude protein, fat content, macro- and micro-elements, and vitamins". Where are the detailed data on the 17 variables? Please add this information as a Table in SI, so readers know what they were! In the text, you used "protein yield" without presenting nutrition values, why? Please be consistent with the term used in the text! Also, please add discussion points in Discussion section on the observation of the highly significant "nutrition yield and soil health" correlation ($r=0.95^{***}$). Which nutrition variable contributed the most to the nutrition value? Was the protein yield?
 - 24) Line 768, please correct the statement into: "In a, b, and d, treatment...."
 - 25) Line 482, please change 'during' into 'at',... During is a period of time/days/weeks/months etc.
 - 26) Line 505, there is no Table S7 in the SI file. It should be Table S6.
 - 27) Lines 337-341, please indicate when these measurements were taken? In Oct 2016 or at the end of the experiment in 2022?
 - 28) Line 349, note the use of the parathesis!
 - 29) Line 349, "These were combined in" not clear, please change to: "There were five newly designed rotations....."
 - 30) Lines 357-358. "Unlike pure WM, the alternative rotations included fallow 358 seasons for parts of their non-WM years." Please cite Fig S1 here.
 - 31) Line 367-368, "Later fertilizer topdressing was performed according to each crop's key growth stages". Please change this into "Topdressing of N nutrient was performed at the key growth stages with specific amounts summarized in Table S2."
 - 32) Line 384, β_i should be protein concentration (%), not content.
 - 33) Line 385-386, please write: For the calculation of economic benefits, the following exchange rate was used: \$1 US = 6.95 Chinese Yuan and 1 € = 7.60 386 Yuan as May 2023, and was adjusted using CPI (Table S3).

Reviewer #2:

Remarks to the Author:

General Comments:

This study conducted a six-year field experiment in the North China Plain to investigate the benefits of diversifying traditional cereal monoculture (wheat-maize) with cash crops (sweet potato) and legumes (peanut and soybean). The research quantifies the significant contributions of diversified crop rotations to enhanced crop yields, reduced N₂O emissions, and improved greenhouse gas balance. Furthermore, it investigates the impact on soil microbial activities, increased soil organic carbon stocks, and overall soil health. This study represents an innovative and noteworthy exemplar of sustainable agricultural practices, emphasizing the global importance of crop diversification for agricultural resilience and soil health.

Overall, this analysis is potentially significant and warrants publication with some necessary modifications. Key points to address include:

- 1) Inclusion of discussions regarding policy implications and the feasibility of diversified rotation farming models is strongly recommended to underscore the broader significance of this research.
- 2) Consider a more integrated approach by combining the Discussion section with the Results section and using subheadings to delineate different facets of the study, e.g., "Diversified Crop Rotations Enhance Ecosystem Productivity," "Diversified Crop Rotations Mitigate Net Greenhouse Gas Emissions," and so forth.
- 3) Methods section should explicitly state its limitations to ensure transparency and rigor in the research.

Specific Comments:

Line 37-40: The statement about "The large-scale adoption of diversified cropping systems in the NCP could increase cereal production by 32% when wheat-maize follows alternative crops in rotation and farmer income by 20% while benefiting the environment" lacks corresponding support in the main text. Ensure that every numerical value mentioned in the abstract is explained in the main text.

Line 52: Place the comma outside the brackets.

Line 124/133: Instead of "by 30, 42, and 49%," use "30%, 42%, and 49%."

Line 144: See the previous comment.

Line 161: Correct the misuse of the negative sign.

Line 506: Consider including equations in the Statistical Section for improved readability.

Line 547: Verify the abbreviations used for the journal "Nature Sustainability."

Reviewer #3:

Remarks to the Author:

This is an authoritative and comprehensive analysis of the impacts of diversifying crop rotations on environmental and economic outcomes in the North China Plain. The authors have undertaken of rigorous analysis of multiple parameters over a period of six years in order to derive robust conclusions around the importance of crop diversification on crop yields, greenhouse gas emissions carbon sequestration soil health and microbial activity. Few previous studies have brought together such a large number of indicators over comparable time spans. They have used state-of-the-art approaches to assessment of agronomic, environmental and economic outcomes, and the findings are likely to be of interest to a wide readership.

There are some concerns regarding the analysis and interpretation of data but I would invite the authors to consider:

Line 91. I would suggest that diversified systems reduce damage to the environment and soil ecosystems rather than completely avoid it.

Line 216. It is argued that diversified cropping systems could offset 106.8 million tonnes of CO₂ equivalents annually. It is not clear on what basis these figures are calculated, and given that extrapolation would cover a wide range of soil and climatic conditions, there must be caveats around the uncertainties associated with this prediction.

Line 247. It is suggested that sweet potatoes and legumes increase carbon use efficiency and promote the formation of highly stable mineral associated soil carbon complexes. These statements do not appear to be based on experimental evidence from this study. Please clarify.

Line 263. Reductions in nitrous oxide emission were ascribed to lower C:N ratios of legume residues, however it could be argued that such lower ratios may actually encourage nitrous oxide emissions (<https://doi.org/10.1111/gcb.16962>)

While I would agree that there is novelty in the approach that is being presented in this publication, there are studies that have shown comparable results (<https://doi.org/10.1016/j.agee.2022.108220>)

Manuscript Title: Diversifying crop rotation increases food production while reducing net greenhouse gas emissions and benefiting soil health

Manuscript ID: NCOMMS-23-41606-T

Journal submitted: Nature Communications

Date of comments received: 13 Oct., 2023

Response to Reviewer #1's general and detailed comments

Comment [1]: What are the noteworthy results?

As a reviewer, I find that the results of this six-year field experiment are quite interesting. The most noteworthy results are: (1) more diversified food/crop production systems increased farms' income while not reducing crop yields as compared with less diversified or traditional cropping systems; (2) diversification significantly reduced greenhouse gas emissions due to the inclusion of legume crops in crop rotation reducing the use of N fertilizers; (3) cropping system diversification enhanced soil health due to improved soil microbial activities. These results are novel and informative!

[Response]: Thanks for your positive comment.

Comment [2]: Will the work be of significance to the field and related fields?

Yes, the work is indeed of significance to the related field. I would think that the work, once published, will have a significant impact on policy-making regarding crop diversification and emission mitigation and improvement of soil health.

[Response]: Thanks for your positive comment.

Comment [3]: How does it compare to the established literature? If the work is not original, please provide relevant references

This work is original. Compared with the existing literature in which numerous publications on the similar topic describing the effect of crop and cropping systems diversification on crop yield, pest control, and systems resilience etc, but little information has been documented in regard to how crop diversification can reduce carbon emissions and at the same time enhancing soil health. The latter is a hot topic currently in the world research community.

[Response]: Thanks for your positive comments.

Comment [4]: Does the work support the conclusions and claims, or is additional evidence needed? Are there any flaws in the data analysis, interpretation and conclusions? - Do these prohibit publication or require revision? Is the methodology sound? Does the work meet the expected standards in your field? Is there enough detail provided in the methods for the work to be reproduced?

This paper, as it is in its current form, is not ready to publish yet because I find a number

of errors, unclear statements, and I would suggest authors address them before it is accepted for publication even although they are mostly minor concerns:

Response: Thanks for the comments. Our point-by-point responses to each comment are below.

Comment [5]: Line 35, the statement “improved the system’s greenhouse gas balance by 88%”. Please explain what this meant? Did you really mean improved GHG ‘emission balance’ by 88%? i.e., CO₂ eq emissions: carbon sequestration?

Response: Thanks. “improved the system’s greenhouse gas balance by 88%” means “reduced net greenhouse gas emissions by 88%”. The statement on greenhouse gas balance refers to net greenhouse gas emissions, including indirect emissions from crop inputs and direct N₂O and CH₄ emissions as well as soil carbon sequestration (all expressed as CO₂ equivalents). The GHG balance is improved by 88% as compared to business as usual.

Comment [6]: Line 36, “legume inclusion” not clear to me.

Response: We have revised the text to “*including legumes in crop rotations stimulated soil microbial activities...*”.

Comment [7]: Line 40, “while benefiting the environment”, please more specific, what environmental benefits they offered?

Response: Thanks. The previous sentences outline the environmental benefits, i.e. improved soil health and reduced GHG emissions. Further details are provided in the first paragraph in the revised manuscript.

Comment [8]: Line 58, “double-cropping systems” are not always bad, they can be beneficial in many other regions. Please modify the statement.

Response: Agreed. We have modified the sentence as follows: “*The loss of soil fertility which may go along with the intensification of crop production further complicates...*”.

Comment [9]: Line 65, change “agricultural commodities” to “agricultural products”.

Response: Thanks, changed as “.....—*while providing other agricultural products such as animal feed, industrial fiber, or multi-purpose biofuels.....*”

Comment [10]: Line 68, add a hyphen between legume and diversified

Response: Thanks, the hyphen is added.

Comment [11]: Lines 72-73: please add reference(s) about the statement “the North China Plain (NCP) —the food basket of China—one of the most intensively cultivated regions in the world”

Response: Thanks. The citation has been added at the end of the sentence.

Comment [12]: Lines 67, 78, 81, 722, you have used the following inconsistent terms on: farmer income, farmers income, farms' incomes, or farm incomes! Please be consistent!

Response: Thanks. We decided to use the term “farmers’ income”.

Comment [13]: Line 87, what did ‘environmental cost’ mean?

Response: Thanks. We have changed the sentence to: “and (c) integrating diversified rotations increases food production, reduces GHG emissions and benefits soil health”

Comment [14]: Line 88, change “we concluded that” into “our results demonstrate that

Response: Thanks, changed.

Comment [15]: Fig. 1 caption, please change “diversified rotation systems with cash and legume crops...” into “rotation systems diversified with cash and legume crops...”

Response: Thanks, changed.

Comment [16]: Fig. 2 caption, line 729-730: please change to: “the dash lines in d-f are the baseline values of the WM rotation”.

Response: Thanks, changed.

Comment [16]: Fig. 3 caption, line 739, please change to “e, changes in soil carbon stock in the 0–90 cm depth...”

Response: Thanks, changed.

Comment [17]: Fig. 3, panels d and f, please switch X-axis with Y-axis to make them consistent with panel e. in fact, you can not switch the two axes between the “cause” and the “effect”.

Response: Thanks. We have remodified Fig. 3 in the revised manuscript as follows.

Comment [18]: Fig. 3 panel, not clear to me why it was called “net GHG emissions” as it presents both direct and indirect emissions and soil carbon sequestration. Would be “net” an end product?

Response: Agreed, we have deleted “net”.

Comment [19]: Line 742, delete the words “on each indicator”

Response: Thanks, deleted.

Comment [20]: Line 753, please delete the word “respectively.”

Response: Thanks, deleted.

Comment [21]: Fig 4, panel b, please add a baseline of the WM rotation, like what were done in Fig. 2d-f. Also, please add percentage values in b to indicate the increment from 2016 to 2022, like what were done in Fig. 5, panels a-d.

Response: Thanks, we have modified Fig. 4b as follows:

Comment [22]: Line 760, please correct the statement into: Treatment abbreviations are the same as those in Fig. 2.

Response: Thanks, corrected.

Comment [23]: Fig. 6, panel b. The CEI index calculation using eqs. 15-24 shown in the SI file, where x_{ij} is the average value of the j th index in the i th crop rotation. Not clear to me, average from what? From replicates? If so, please add $n=3$ in the statement, if not, please tell readers how it was averaged.

Response: Thanks. X_{ij} is the average value of measured replications ($n=3$). We have added this information to the revised SI file.

Comment [24] The calculation of CEI, as described in lines 131-170 of the SI file, please provide more details about how the CEI was processed and using what software program?

Response: Thanks. Lines 131-170 in the SI show the process used for calculating CEI,

which is based on Entropy-TOPSIS^[14-16]. The calculation was done in an Excel sheet following Equations 15–24 in the SI file. Graphics were generated using Origin 2021 software.

References:

14. Wang, H. D. *et al.* Multi-objective optimization of water and fertilizer management for potato production in sandy areas of northern China based on TOPSIS. *Field Crops Res.* **240**, 55-68 (2019).
15. Chen, Y., Zhu, M. K., Lu, J. L., Zhou, Q. & Ma, W. B. Evaluation of ecological city and analysis of obstacle factors under the background of high-quality development: Taking cities in the Yellow River Basin as examples. *Ecol. Indic.* **118**, 106771 (2020).
16. Liu, X. G., Peng, Y. L., Yang, Q. L., Wang, X. K. & Cui, N. B. Determining optimal deficit irrigation and fertilization to increase mango yield, quality, and WUE in a dry hot environment based on TOPSIS. *Agric. Water Manage.* **245**, 106650 (2021).

Comment [25] Fig. 6, panel c, stated that “Nutrition yield includes 17 nutritional components of each crop such as crude protein, fat content, macro- and micro-elements, and vitamins”. Where are the detailed data on the 17 variables? Please add this information as a Table in SI, so readers know what they were! In the text, you used “protein yield” without presenting nutrition values, why? Please be consistent with the term used in the text! Also, please add discussion points in Discussion section on the observation of the highly significant “nutrition yield and soil health” correlation ($r=0.95^{***}$). Which nutrition variable contributed the most to the nutrition value? Was the protein yield?

Response: Thanks. The 17 nutritional components for the different crops have been listed in Table S4 of the original SI as follows:

“In addition to harvest weight and volume, we also determined the nutritional value of harvested crops based on their nutrient composition² (Table S4).”

Reference:

9. Groot, J. C. J. & Yang, X. L. Trade-offs in the design of sustainable cropping systems at a regional level: A case study on the North China Plain. *FASE* **9**, 14 (2022).

In the text we used “Protein yield” instead of nutritional value as “Protein yield” was the dominating factor to the nutrition value, therefore, in Fig. 1 d–f, we compared grain yield (Fig.1d), economic benefit (Fig.1e), and protein yield (Fig.1e) of the succeeding cereal crops. For the comprehensive evaluation index, we used nutrition yield calculated as nutrition score (described in line 76-100 in SI file), including 17 nutritional components for each crop (e.g., crude protein, fat content, macro- and micro-elements, and vitamins).

We have also discussed the correlation between nutrition yield and soil health in the

Discussion:

“Furthermore, our findings highlight a close correlation between soil health and nutrition yield, underscoring the interconnectedness of both dimensions and their reliance on agricultural practices³⁹”.

Reference:

39. Montgomery, D. R. & Biklé, A. Soil Health and Nutrient Density: Beyond Organic vs. Conventional Farming. *Front. Sustainable Food Syst.* 5, 417 (2021).

Comment [26] Line 768, please correct the statement into: “In a, b, and d, treatment....”

Response: Thanks, corrected.

Comment [27] Line 482, please change ‘during’ into ‘at’,... During is a period of time/days/weeks/months etc.

Response: Thanks, changed. “.....and at the final summer maize harvest.....”

Comment [28] Line 505, there is no Table S7 in the SI file. It should be Table S6.

Response: Thanks, corrected. “.....and w is the weighting factor for each soil indicator (Table S6).....”

Comment [29] Lines 337-341, please indicate when these measurements were taken? In Oct 2016 or at the end of the experiment in 2022?

Response: Thanks. They are the basic soil properties, measured before the experiment started in Oct 2016.

“.....The 0–10 cm soil layer before the experiment start has $\text{pH } 7.6 \pm 0.15$”.

Comment [30] Line 349, note the use of the parathesis!

Response: Thanks, corrected.

Comment [31] Line 349, “These were combined in” not clear, please change to: “There were five newly designed rotations.....”

Response: Thanks, done.

Comment [32] Lines 357-358. “Unlike pure WM, the alternative rotations included fallow seasons for parts of their non-WM years.” Please cite Fig S1 here.

Response: Thanks, it is cited.

Comment [33] Line 367-368, “Later fertilizer topdressing was performed according to each crop’s key growth stages”. Please change this into “Topdressing of N nutrient was performed at the key growth stages with specific amounts summarized in Table S2.”

Response: Thanks, changed.

Comment [34] Line 384, β_i should be protein concentration (%), not content.

Response: Thanks, corrected.

Comment [35] Line 385-386, please write: For the calculation of economic benefits, the following exchange rate was used: \$1 US = 6.95 Chinese Yuan and 1 € = 7.60 386 Yuan as May 2023, and was adjusted using CPI (Table S3).

Response: Thanks, done.

Response to Reviewer #2's general and detailed comments

Comment [1] This study conducted a six-year field experiment in the North China Plain to investigate the benefits of diversifying traditional cereal monoculture (wheat-maize) with cash crops (sweet potato) and legumes (peanut and soybean). The research quantifies the significant contributions of diversified crop rotations to enhanced crop yields, reduced N₂O emissions, and improved greenhouse gas balance. Furthermore, it investigates the impact on soil microbial activities, increased soil organic carbon stocks, and overall soil health. This study represents an innovative and noteworthy exemplar of sustainable agricultural practices, emphasizing the global importance of crop diversification for agricultural resilience and soil health.

Response: Thanks for your positive comments.

Overall, this analysis is potentially significant and warrants publication with some necessary modifications. Key points to address include:

Comment [2] Inclusion of discussions regarding policy implications and the feasibility of diversified rotation farming models is strongly recommended to underscore the broader significance of this research.

Response: Thanks. We have reflected on these very important points and revised the last paragraph of the discussion section accordingly:

“The novelty of this study is its highly diversified crop rotations, with cash crops and legumes replacing cereal monocultures, together with rotating shallow-root crops with deep-rooted crops. It positively affects ecosystem services and functions, partly compensating for the lower services in less diversified cereal-based rotations. Crop diversification offers significant benefits in reducing GHG emissions and has a synergistic effect on plant biomass and protein production, soil health, and microbial community biodiversity. Diversification reduced crop inputs, particularly N fertilizer and irrigation amounts, minimizing environmental costs while increasing farmers' incomes. Thus, diversified crop rotations can serve as an effective strategy for sustainable food production in areas worldwide with environments similar to the NCP.

The scientific evidence presented in this study holds significant potential for informing and reinforcing current policies and incentive schemes to improve the green transition of farming systems in China. Specifically, China set objectives to promote soybean production (initiated in 2019), implement an action plan for zero increase in fertilizer and pesticide use (launched in 2015), and develop strategies to reduce agricultural GHG emissions with the ultimate goal of transforming agricultural systems into zero ‘emitters’ by 2060. When designing effective

diversified cropping systems, it is important to consider local environmental conditions to balance land use for food production with the other functions of an ecosystem. Optimizing crop configuration is essential—through social, economic, and environmental dimensional planning within a complex framework of various indicators to ensure long-term sustainability. We recommend that developing and adopting diversified cropping systems should be a key consideration in agricultural policy setting and a top priority for on-farm decision-making, as they are critical for achieving long-term food production resilience, maintaining soil health, and ensuring environmental sustainability. Given that climate variability will increase, multi-year assessments of the sustainability of agricultural production systems remain tentative, and funding should be sought for additional observation sites for long-term monitoring.”

Comment [3] Consider a more integrated approach by combining the Discussion section with the Results section and using subheadings to delineate different facets of the study, e.g., "Diversified Crop Rotations Enhance Ecosystem Productivity," "Diversified Crop Rotations Mitigate Net Greenhouse Gas Emissions," and so forth.

Response: Thanks. We have closely followed the guidelines provided by the journal. We prefer to keep the Results and Discussion sections separate, as recommend by the journal.

Comment [4] Methods section should explicitly state its limitations to ensure transparency and rigor in the research.

Response: Thanks. We added the following paragraph at the end of the Methods section to reflect the limitations.

“The results presented in the paper were derived predominantly from a six-year field experiment undertaken at the Luancheng research station, a typical representative site for production systems and environmental characteristics in the NCP. Multi- location experiments will ensure the spatiotemporal effects of diversified crop rotations on system productivity, environmental impacts, and socioeconomic performance. Moreover, inter-annual weather conditions are highly variable, and this may intensify in the future. Thus, longer term monitoring at multiple sites would be valuable for reducing the uncertainty associated with extreme weather events and climate change.”

In addition, we have also added the limitation to the last part of the Discussion, with details in the citation provided in response to comment [2].

Comment [5] Line 37-40: The statement about “The large-scale adoption of diversified cropping systems in the NCP could increase cereal production by 32% when wheat-maize follows alternative crops in rotation and farmer income by 20% while benefiting the environment” lacks corresponding support in the main text. Ensure that every numerical value mentioned in the abstract is explained in the main text.

Response: Thanks. We have provided each number in the first paragraph of the discussion section:farmers income by 20% in Line 228;increase cereal production by 32% in Line235 in the revised manuscript.

“.....Farmers in the NCP region could benefit from 20% increases in annual net income, equivalent to 84 billion Yuan (\$11.6 US billion) in total.....”

“.....Furthermore, wheat and maize yields may increase by 32%, equivalent to 73.5 million tonnes per year if planted following alternative crops such as sweet potato, soybean or peanuts; this would make about 36.1 million tonnes of additional straw biomass available annually for alternative uses, such as feed, bioenergy, or enhancing soil carbon stocks.....”

Comment [6] Line 52: Place the comma outside the brackets.

Response: Thanks, changed.

Comment [7] Line 124/133: Instead of "by 30, 42, and 49%," use "30%, 42%, and 49%."

Response: Thanks, changed.

Comment [8] Line 144: See the previous comment.

Response: Thanks, changed.

Comment [9] Line 161: Correct the misuse of the negative sign.

Response: Thanks, corrected.

Comment [10] Line 506: Consider including equations in the Statistical Section for improved readability.

Response: Thanks. We have used standard statistical methods and R packages which are detailed in the link (<https://figshare.com/s/5c07d2ac7c8dc4a5b678>).

Comment [11] Line 547: Verify the abbreviations used for the journal "Nature Sustainability."

Response: Thanks, corrected to "Nat. Sustain."

Response to Reviewer #3's general and detailed comments

Comment [1] This is an authoritative and comprehensive analysis of the impacts of diversifying crop rotations on environmental and economic outcomes in the North China Plain. The authors have undertaken of rigorous analysis of multiple parameters over a period of six years in order to derive robust conclusions around the importance of crop diversification on crop yields, greenhouse gas emissions carbon sequestration soil health and microbial activity. Few previous studies have brought together such a large number of indicators over comparable time spans. They have used state-of-the-art approaches to assessment of agronomic, environmental and economic outcomes, and the findings are likely to be of interest to a wide readership.

Response: Thanks for your positive comments.

There are some concerns regarding the analysis and interpretation of data but I would invite the authors to consider:

Comment [2] Line 91. I would suggest that diversified systems reduce damage to the environment and soil ecosystems rather than completely avoid it.

Response: Agreed. We have modified it in the revised manuscript.

“(a) instructive findings from newly designed, tested, and validated diversified systems could guide the NCP in establishing a more sustainable system to maintain or increase grain and protein production with reducing the damage to the environment and soil ecosystems”

Comment [3] Line 216. It is argued that diversified cropping systems could offset 106.8 million tonnes of CO₂ equivalents annually. It is not clear on what basis these figures are calculated, and given that extrapolation would cover a wide range of soil and climatic conditions, there must be caveats around the uncertainties associated with this prediction.

Response: Thanks. The calculation of the carbon benefit was based on a simple linear extrapolation using the reduction in net GHG emissions of diversified crop rotations of $6.1 \pm 1.8 \text{ t CO}_2\text{-eq ha}^{-1} \text{ yr}^{-1}$ on average (compared to conventional WM rotation) multiplied by the average sown area of double-season cropping systems in the North China Plain ($17.6 \times 10^6 \text{ ha yr}^{-1}$) (Liu et al., 2022), resulting in the estimated 106.8 ± 31.7 million tonnes of CO₂ equivalents. While soil properties are primarily comparable across the NCP regarding texture and SOC content (Han et al., 2018), climatic conditions may differ somehow in temperature but are relatively minor in rainfall (summer monsoon-driven rainfall systems). Therefore, we are confident that our estimate of the net GHG emission reduction is robust. To provide an uncertainty range of our estimate, we used the variability of reductions in net GHG emissions observed

in our experiments, i.e., 1.8 t CO₂-eq ha⁻¹ yr⁻¹.

References:

Liu, Z. J. et al. Patterns and causes of winter wheat and summer maize rotation area change over the North China Plain. *Environ. Res. Lett.* **17**, 044056 (2022).

Han, D. R. et al. Large soil organic carbon increase due to improved agronomic management in the North China Plain from 1980s to 2010s. *Global Change Biol.* **24**, 987-1000 (2018).

Comment [4] Line 247. It is suggested that sweet potatoes and legumes increase carbon use efficiency and promote the formation of highly stable mineral associated soil carbon complexes. These statements do not appear to be based on experimental evidence from this study. Please clarify.

Response: Thanks. We have revised the sentence to clarify that these findings were made in previous studies.

“Introducing sweet potatoes and legumes into crop rotations has been shown to stimulate microbial growth, increase C use efficiency³⁶, and to promote the formation of highly stable mineral-associated soil C complexes³⁷.”

References:

36. Liu, K., Bandara, M., Hamel, C., Knight, J. D. & Gan, Y. T. Intensifying crop rotations with pulse crops enhances system productivity and soil organic carbon in semi-arid environments. *Field Crops Res.* **248**, 107657 (2020).

37. Chenu, C. et al. Increasing organic stocks in agricultural soils: Knowledge gaps and potential innovations. *Soil Tillage Res.* **188**, 41-52 (2019).

Comment [5] Line 263. Reductions in nitrous oxide emission were ascribed to lower C:N ratios of legume residues, however it could be argued that such lower ratios may actually encourage nitrous oxide emissions (<https://doi.org/10.1111/gcb.16962>)

Response: Thanks. We have revised the Discussion in the revised manuscript. We agree that legumes, specifically incorporating legume litter into soil in most cases, increases soil N₂O emissions. However, a legume crop in the crop rotation reduces the total amount of fertilizer N used within a cropping period and, thus, the total N₂O emissions across the cropping period. We have revised the sentence as follows to reflect our thoughts:

“Legume crops improve soil N cycling⁴⁰ but may also increase N₂O emissions, especially following residue incorporation^{41,42}. However, legume cultivation requires significantly less fertilizer than cereal crops such as maize or wheat. Thus, the PWM and SWM crop rotations required 37% less fertilizer than conventional WM, explaining the overall reduction in N₂O emissions for the legume-based rotations as N fertilizers were the main source of N₂O emissions in our study⁴⁰.”

References:

40. Guinet, M., Nicolardot, B. & Voisin, A.-S. Nitrogen benefits of ten legume pre-crops for wheat assessed by field measurements and modelling. *Eur. J. Agron.* **120**, 126151 (2020).
41. Abdalla, M. et al. A critical review of the impacts of cover crops on nitrogen leaching, net greenhouse gas balance and crop productivity. *Global Change Biol.* **25**, 2530-2543 (2019).
42. Olesen, J. E. et al. Challenges of accounting nitrous oxide emissions from agricultural crop residues. *Global Change Biol.* **00**, 1-10 (2023).

Comment [6] While I would agree that there is novelty in the approach that is being presented in this publication, there are studies that have shown comparable results (<https://doi.org/10.1016/j.agee.2022.108220>)

Response: Thanks for your positive comment. We have cited this reference in our manuscript for comparable results.

Reviewers' Comments:

Reviewer #1:

Remarks to the Author:

All my comments were taken into account in a satisfactory manner. I have no further remark.

Reviewer #2:

Remarks to the Author:

All my comments are well addressed, and I have no more comments.

Reviewer #3:

Remarks to the Author:

he study provides robust evidence for the benefits of diversified crop rotations in delivering reduced greenhouse gas emissions and improved soil health in arable rotations in the North China Plain. The study is based on a synthesis of long term experimental data, and makes an important contribution to our understanding of the contribution of such systems to economic and environmental outcomes. I am satisfied with the modifications to the manuscript following the initial review.